# Functional plasticity in chromosome–microtubule coupling on the evolutionary time scale

Sundar Ram Sankaranarayanan[1], Satya Dev Polisetty[1], Kuladeep Das[1], Arti Dumbrepatil[1], Bethan Medina-Pritchard[2], Martin Singleton[2], A Arockia Jeyaprakash[2,3], Kaustuv Sanyal[1]

The Dam1 complex is essential for mitotic progression across evolutionarily divergent fungi. Upon analyzing amino acid (aa) sequences of Dad2, a Dam1 complex subunit, we identified a conserved 10-aa–long Dad2 signature sequence (DSS). An arginine residue (R126) in the DSS is essential for viability in *Saccharomyces cerevisiae* that possesses point centromeres. The corresponding arginine residues are functionally important but not essential for viability in *Candida albicans* and *Cryptococcus neoformans*; both carry several kilobases long regional centromeres. The purified recombinant Dam1 complex containing either Dad2$^{\Delta DSS}$ or Dad2$^{R126A}$ failed to bind microtubules (MTs) or form any visible rings like the WT complex. Intriguingly, functional analysis revealed that the requirement of the conserved arginine residue for chromosome biorientation and mitotic progression reduced with increasing centromere length. We propose that plasticity of the invariant arginine of Dad2 in organisms with regional centromeres is achieved by conditional elevation of the kinetochore protein(s) to enable multiple kinetochore MTs to bind to each chromosome. The capacity of a chromosome to bind multiple kinetochore MTs may mask the deleterious effects of such lethal mutations.

## Introduction

Accurate segregation of duplicated genetic material requires dynamic and regulated attachment of the sister chromatids to spindle microtubules (MTs). This association is mediated by the kinetochore, a multiprotein complex that assembles on the centromeric locus on every chromosome (Cheeseman, 2014). The centromere-specific histone H3 variant, CENPA (Cse4 in yeast), is a hallmark of functional centromeres and acts as the foundation for assembling other kinetochore proteins in most organisms (Cheeseman, 2014; Musacchio & Desai, 2017). The kinetochore ensemble spans 125 bp

on each point centromeres of *Saccharomyces cerevisiae* chromosomes and forms a single MT-binding module. Kinetochores on the longer regional centromeres of most other fungi and metazoans are considered as an array of the module formed on the point centromeres (Joglekar et al, 2006, 2008; Weir et al, 2016; Musacchio & Desai, 2017; Walstein et al, 2021). This enables the large regional centromeres to support the binding of multiple MTs on each chromosome instead of a single MT binding to an *S. cerevisiae* chromosome (Ding et al, 1993; Winey et al, 1995).

The primary MT-binding module of the kinetochore ensemble is the Ndc80 complex. However, the load-bearing ability of the Ndc80 complex under tension is augmented by a fungal-specific Dam1 complex of the outer kinetochore (Lampert et al, 2010, 2013; Tien et al, 2010). This heterodecameric complex, comprising of Dad1, Dad2, Dad3, Dad4, Dam1, Duo1, Ask1, Hsk3, Spc19, and Spc34 subunits, localizes explicitly to the kinetochore in an Ndc80-dependent manner and is essential for sister kinetochores to establish and maintain a bioriented state until anaphase onset (Hofmann et al, 1998; Cheeseman et al, 2001a, 2001b; Enquist-Newman et al, 2001; Janke et al, 2002; Li et al, 2002). They accomplish this by their ability to track and sustain attachment with the depolymerizing MT ends. Consistently, mutants of the Dam1 complex subunits show severe chromosome segregation defects in all organisms studied to date (Hofmann et al, 1998; Enquist-Newman et al, 2001; Janke et al, 2002; Liu et al, 2005; Thakur & Sanyal, 2011; Chatterjee et al, 2016; Sridhar et al, 2021). They also found essential for viability in organisms with point or short regional centromeres that bind to only one MT on each chromosome (as in *S. cerevisiae* and *Candida albicans*) and those harboring longer centromeric chromatin like the basidiomycete *Cryptococcus neoformans* (Yadav et al, 2018; Sridhar et al, 2021). The notable exceptions to this are *Schizosaccharomycces pombe* and *Magnaporthe orzyae* known to have long regional centromeres (Liu et al, 2005; Shah et al, 2019).

Analysis of in vitro purified Dam1 complex from *S. cerevisiae* and *Chaetomium thermophilum* revealed that the heterodecameric Dam1 complex oligomerized (17-mer) to form a ring-like structure around the MTs (Miranda et al, 2005; Westermann et al, 2005; Jenni &

---

[1]Molecular Mycology Laboratory, Molecular Biology and Genetics Unit, Jawaharlal Nehru Centre for Advanced Scientific Research, Bengaluru, India [2]Wellcome Centre for Cell Biology, University of Edinburgh, Edinburgh, UK [3]Gene Center and Department of Biochemistry, Ludwig-Maximilian-Universität, Munich, Germany

Correspondence: sanyal@jncasr.ac.in
Sundar Ram Sankaranarayanan's present address is Institut Curie, PSL Research University, Sorbonne Université, CNRS, UMR3664 Nuclear Dynamics, Paris, France

Harrison, 2018). Until now, such structures have been detected in vivo only in *S. cerevisiae* (Ng et al, 2019). The universality of this conformation was contested by observations in *S. pombe* wherein the cellular levels of Dam1 complex subunits were found insufficient to adopt a ring-like structure (Joglekar et al, 2008; Gao et al, 2010). They were proposed to form smaller oligomeric patches (3–5mer) along the MTs. The cellular cues that determine the physiological state of Dam1 complex in each species remain unclear.

The kinetochore–MT interface exhibits a remarkable plasticity in terms of their composition and interactions. This structural plasticity of a kinetochore plays a significant role in maintaining dynamic interactions with the associated kinetochore MTs (kMTs) across the mitotic cell cycle. A well-known example is the presence of multiple linker pathways connecting the inner and outer kinetochores and to temporally switch between these pathways in a cell cycle-dependent manner. (Milks et al, 2009; Schleiffer et al, 2012; Hornung et al, 2014; Dimitrova et al, 2016; Sridhar et al, 2021). Proteins of the outer kinetochore like Ndc80 are enriched further at anaphase once a minimal kinetochore assembles to satisfy the mitotic checkpoint. This highlights the compositional variations at the kinetochore–MT interface (Malvezzi et al, 2013; Dhatchinamoorthy et al, 2017, 2019). Studies in engineered cells revealed the ability to seed and form additional kMTs beyond the number observed in a wild-type (WT) state. For instance, over 50 copies of centromeric plasmids could be tolerated in *S. cerevisiae* as additional MTs formed to bind to the kinetochores assembled on these plasmids (Nannas et al, 2014). In another ascomycete, *C. albicans*, overexpression of CENPA resulted in the recruitment of excess outer kinetochore subunits on each centromere (Burrack et al, 2011). These molecules now act as receptors to bind multiple kMTs on each chromosome as opposed to a single kMT binding to each 3–5 kb long regional centromere in a WT *C. albicans* cell. Functional benefits of this plasticity at the kinetochore are less explored in the context of survival/fitness advantage in conditions that are non-conducive for chromosome segregation.

In this study, we explore the functional plasticity of the kinetochore–MT coupler in fungi as a function of tolerance to substitution of an evolutionarily conserved arginine residue in Dad2, a Dam1 complex subunit. Genetic analysis reveals that the conserved arginine residue in Dad2 is essential for viability and is required for chromosome biorientation and mitotic progression in *S. cerevisiae*. Intriguingly, a similar substitution of the corresponding arginine residue in organisms with longer centromeric chromatin could be tolerated to a varying extent.

# Results

## The Dad2 signature sequence (DSS) is a conserved motif at the C-terminus in the Dad2 protein family

The Dad2 protein family is typically defined by the presence of the Dad2 domain (PF08654), a conserved amino acid sequence stretch at the N-terminus (Fig 1A, *top*). To further understand the domain architecture, we analyzed the primary amino acid sequence of Dad2 from more

than 500 fungal species collectively representing the three major fungal phyla—Ascomycota, Basidiomycota, and Mucoromycota (See the Materials and Methods section). We identified a 10-amino acid-long evolutionarily conserved sequence motif towards the C- terminus of Dad2 and named it as the DSS (Fig 1A). The DSS motif remains conserved across species with diverse centromere structures known among fungi. Although the extent of conservation of most amino acid residues is variable, the DSS possesses an almost invariant arginine residue (R') (Fig 1A). Further analysis of the DSS within specific fungal phyla revealed other features associated with this motif. Among the members of Saccharomycotina, the second position from the invariant arginine was also encoded by a positively charged amino acid such as arginine (R) or lysine (K) in low frequency. However, this position was predominantly represented by proline (P) in other fungal orders such as Pezizomycotina, Taphrinomycotina, and in the fungal phylum of Basidiomycota. Across species, we find the conserved arginine R' centered on a short hydrophobic patch. Interestingly, in the available three-dimensional structure of the Dam1 complex (PDB:6CFZ) (Jenni & Harrison, 2018), the DSS motif makes multipartite interactions with residues of Dam1, Spc19, and Spc34 subunits within the "central domain," suggesting its potential contribution to the overall structure and function of the Dam1 complex (Fig 1B). Particularly, the conserved arginine residue makes both hydrophobic and electrostatic interactions with residues of Spc19 which were also conserved across several fungi (Fig S1). Considering the extent of conservation of the DSS across the three major fungal phyla, we sought to understand the functional significance of this domain in Dad2, especially its conserved arginine residue in two fungal systems —*S. cerevisiae* and *C. albicans*, diverged from each other by 117 mya (Shen et al, 2018), and known to possess distinct centromere structures and kinetochore architectures.

## The essentiality of the DSS is not a conserved feature across centromere lengths

To study the DSS function in *S. cerevisiae*, we generated constructs expressing mutant versions of Dad2 bearing alanine substitutions of the conserved arginine residues or a truncated version lacking the entire DSS domain (ScDad2-R126A, ScDad2-R128A, and ScDad2-ΔDSS; Fig 1C). To test the function of these mutants, we used a previously reported temperature-sensitive (*ts*) mutant of *DAD2* wherein the genomic allele of *DAD2* was deleted and a *ts* allele was reintegrated (CJY077, *dad2::KanMX6 his3Δ200 leu2Δ1::pCJ055 [dad2^ts]*) (Janke et al, 2002). This mutant strain was independently transformed with a pRS313-based *CEN/ARS* plasmid containing either WT or mutant versions of *DAD2* (as in Fig 1C) tagged with GFP at the C-terminus. In all these strains, the expression of *DAD2-GFP* was driven by the native promoter of Sc*DAD2*. The functional significance of each of these residues was tested by a spot dilution assay to determine the ability of the mutants to support growth at an elevated temperature of 37°C (Fig 1D). As expected, the strain YSR01 carrying the empty vector was unable to grow at this temperature, but the strain YSR02 expressing ScDad2-FL grew well. Strikingly, the strains YSR03 expressing ScDad2-R126A and YSR05 expressing ScDad2-ΔDSS failed to grow at 37°C, revealing that R126 makes the

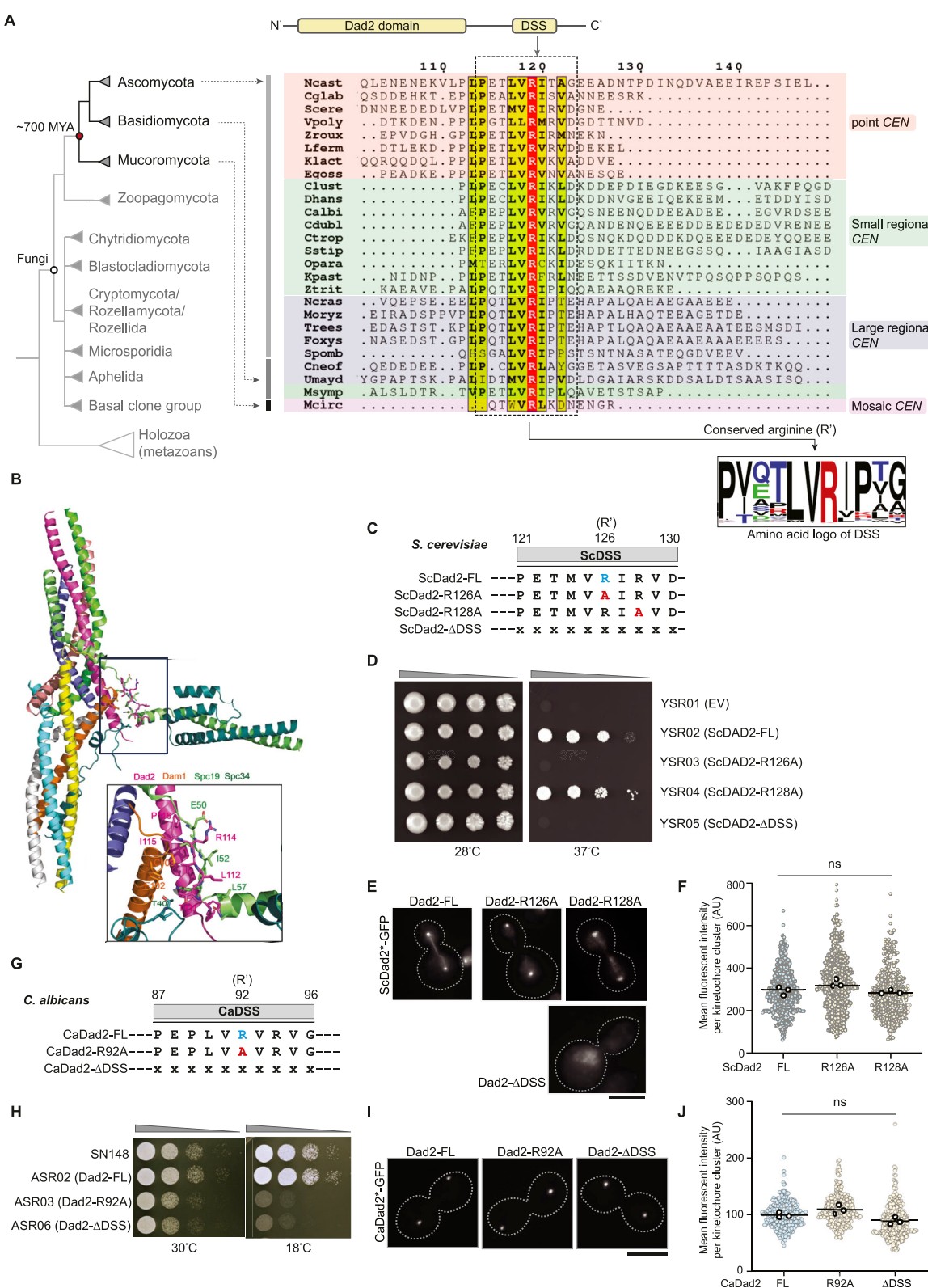

**Figure 1. The functional significance of the conserved Dad2 signature sequence (DSS) motif is dependent on the length of centromeric chromatin.**
**(A)** *Top*, schematic of the Dad2 protein family indicating the known Dad2 domain (PF08654) at the N-terminus and a previously unknown conserved motif at the C-terminus. *Bottom*, the cladogram represents the relative phylogenetic position of major fungal phyla such as Ascomycota, Basidiomycota, and Mucoromycota and the corresponding species belong to them. The time of divergence of Mucoromycota from the common ancestor it shared with Ascomycota and Basidiomycota is based on Berbee et al (2017). A representative alignment of Dad2 amino acid sequences highlighting the conservation of the DSS and its conserved arginine residue (R') across

DSS essential for Dad2 function in *S. cerevisiae*. However, the strain YSR04 expressing ScDad2-R128A also supported growth at an elevated temperature, suggestive of a nonessential role of R128 for viability. To rule out the lack of expression of these mutant proteins as a cause for their inability to complement growth at an elevated temperature, we confirmed the localization of the ectopic Dad2 protein (WT, mutant or truncated) at this temperature (Fig 1E). All versions of Dad2 except ScDad2-ΔDSS showed detectable punctate localization signals typical of clustered budding yeast kinetochores and their localization signal intensities were comparable (Fig 1F) (Cheeseman et al, 2001b; Janke et al, 2002). These results strongly indicated that the entire DSS motif is essential for the kinetochore localization of ScDad2. Based on the location of the DSS in the Dam1 complex monomer, we suspect that the deletion of the DSS domain might affect the integrity of the complex and, in turn, the localization of Dad2.

To rule out the temperature sensitivity of mutant cells expressing ScDad2-R126A and ScDad2-ΔDSS observed in the above assay, we tested the essentiality of this residue at an ambient growth temperature (30°C) as well. A tester strain was engineered such that the genomic allele of Dad2 was deleted after expressing Dad2 from an ectopic allele cloned in a centromeric plasmid carrying the *URA3* gene (Fig S2A). This strain was then transformed with pRS313-based plasmids containing WT or mutant versions of *DAD2*. The ability of strains to lose the protection allele present in the plasmid and support subsequent growth on media containing 5′FOA was used to test for the essentiality of the mutant Dad2 protein for viability. The inability of YSR10 expressing ScDad2-R126A and YSR12 expressing ScDad2-ΔDSS to grow on the Ura⁺ counter-selection media containing 5′FOA at various dilutions tested in the spot dilution assay confirmed the essentiality of R126 and thereby the role of the DSS in cell viability (Fig S2A). These results confirmed that the highly conserved R126 residue is a critical amino acid for the essential function performed by Dad2 in *S. cerevisiae*.

Because ScDad2-R126A and ScDad2-ΔDSS mutants were inviable in *S. cerevisiae*, we sought to test the importance of the corresponding arginine residues in Dad2 that is essential for viability in *C. albicans*. To test this in *C. albicans*, we used a previously reported conditional mutant of *dad2* (J108, *dad2/PCK1pr-DAD2*) (Thakur &

Sanyal, 2011). In this conditional mutant, the only genomic allele of Ca*DAD2* is expressed under the *PCK1* promoter that shuts down its expression in the presence of dextrose in growth media (Fig S2B). Whereas the conditional mutant J108 was unable to grow in media supplemented with dextrose, the complemented strains J108A and J108B expressing CaDad2-FL and CaDad2-ΔDSS, respectively, supported growth in this media, suggesting that unlike in *S. cerevisiae*, the deletion of the DSS motif does not make *C. albicans* cells inviable (Fig S2C). We further validated this observation by engineering the only allele of *DAD2* in ASR01 (*dad2/DAD2*) to express GFP-tagged versions of CaDad2-FL, -R92A, and -ΔDSS in strains ASR02, ASR03, and ASR04, respectively (Fig 1G). Spot dilution assays were performed to assess if any of the CaDad2 derivatives exhibited growth defects as compared with the WT strain. Whereas we could observe a mild growth retardation at 30°C, the growth of mutant strains ASR03 and ASR04 expressing CaDad2-R92A and CaDad2-ΔDSS, respectively, was significantly compromised at a lower temperature of 18°C when compared with CaDad2-FL-expressing strain ASR02 or the WT strain SN148 grown under similar conditions (Fig 1H). We could detect punctate localization signals in strains expressing not only CaDad2-R92A but also CaDad2-ΔDSS, suggesting that kinetochore localization was neither affected by alanine substitution of the critical arginine residue nor when the entire DSS was deleted in *C. albicans* (Fig 1I and J). These experiments reveal that the presence of the evolutionarily conserved arginine residue in Dad2 is dispensable for the viability of *C. albicans* under normal growth conditions. Having observed such phenotypic differences in mutants of Dad2 in *S. cerevisiae* and *C. albicans*, we sought to dissect the mutant phenotype further to study the extent of functional conservation of the DSS between them.

### The arginine residue R126 in the conserved DSS motif is critical for proper spindle dynamics and bipolar attachment of kinetochores in *S. cerevisiae*

To probe deeper into the role of the DSS and its conserved arginine R126 in chromosome segregation in *S. cerevisiae*, we generated

species with known centromere structures is shown. The residue numbers in the top is based on *Naumovozyma castellii* Dad2 sequence. The consensus sequence of the DSS represented as an amino acid logo was generated from a multiple-sequence alignment of 466 Dad2 sequences (Table S1). The height of each amino acid indicates its probability of occurrence at the specific position. Amino acids are color-coded based on the inherent charge contributed by the side chain: black, neutral; red, positive; green, negative; blue, polar. **(B)** Cartoon representation of the three-dimensional structure of the DASH/Dam1 complex from *Chaetomium thermophilum* determined by Jenni & Harrison (2018) (PDB: 6CFZ). The inset shows the close-up views of the Dad2 DSS motif and segments of Dam1, Spc19, and Spc34 which are in close proximity to the DSS motif. Amino acid residues contacting the DSS motif are shown in stick representation. **(C)** The amino acid sequence of the DSS in *S. cerevisiae* Dad2 (ScDad2) depicting the conserved arginine residue (blue) is shown along with the mutants generated to study the function of the DSS in *S. cerevisiae*. **(B, D)** The full-length WT *DAD2* or with mutations in the DSS (121–130 aa in ScDad2) or a truncated version of *DAD2* lacking the DSS as mentioned in (B) were cloned into pRS313. These constructs were used to transform a temperature-sensitive *dad2* mutant CJY077 to express them as C-terminally GFP-tagged proteins. YSR01 corresponds to a control strain carrying the empty vector. Single colonies of the strains YSR01 through YSR05 were grown on CM-Leu-His media for 14 h at 26°C, serially diluted 10-fold, and spotted (10⁵ to 10²) on CM-Leu-His plates and incubated at 26°C and 37°C, respectively. Plates were photographed after 72 h of incubation. **(E)** Fluorescence microscopic images showing the localization of indicated versions of Dad2-GFP in cells grown for 4 h at 37°C. Scale bar, 5 μm. **(F)** The localization efficiency of indicated versions of ScDad2 were compared by measuring the mean fluorescence intensity per kinetochore cluster (*y*-axis) for each of them. The scatter plot in the background represents the distribution of fluorescence intensities. The three black circles in the foreground represent the mean value from each replicate. No statistically significant difference was observed (*t* test with Welsch's correction, n > 100 cells per replicate). Strains expressing ScDad2-DSS was excluded from quantification as they did not show punctate localization. **(G)** The DSS sequences in the indicated strains generated to assay for the role of the DSS and the conserved arginine residue (R′) in *C. albicans*. x indicates the absence of an amino acid residue. **(H)** Spot dilution assay performed with the indicated strains grown in YPDU for 14 h at 30°C. Cells were serially diluted 10-fold, spotted on YPDU plates (10⁵ to 10² cells), incubated at 30°C and 18°C, and imaged after 36 h and 48 h, respectively. **(I)** Fluorescence microscopy images of *C. albicans* cells expressing the indicated versions of Dad2-GFP after logarithmic growth in YPDU at 30°C. Scale bar, 5 μm. **(J)** The localization efficiency of indicated versions of CaDad2 was compared by measuring the mean fluorescence intensity per kinetochore cluster (*y*-axis) for each of them. The scatter plot in the background represents the distribution of fluorescence intensities. The three black circles in the foreground represent the mean value from each replicate. No statistically significant difference was observed (*t* test with Welsch's correction, n > 100 cells per replicate).

conditional mutants of *DAD2* by replacing the endogenous *DAD2* promoter with the *GAL₁₋₁₀* promoter in the strain SBY12503 where spindle pole bodies (SPBs) (Spc110 is tagged with mCherry) and *CEN3* are marked (by the binding of GFP-LacI to LacO arrays integrated adjacent to *CEN3*) (Umbreit et al, 2014). The resulting strain YSR13 (*GALpr-DAD2*) fails to grow when dextrose is the sole carbon source in the growth media (Fig S3A and B). This conditional mutant strain YSR13 was then independently transformed with vectors that reintegrate WT or mutant versions of *DAD2* expressed from the *DAD2* promoter at the native locus. In line with our previous observations, the conditional mutant complemented with ScDad2-FL in YSR14 and ScDad2-R128A in YSR16 supported growth on dextrose-containing media. On the other hand, the strains YSR15 and YSR17 expressing ScDad2-R126A and ScDad2-ΔDSS, respectively, could not complement function as they were unable to support the growth of the conditional mutant on this media (Fig S3B).

To identify the defects that led to viability loss, the *dad2* conditional mutant YSR13 along with the reintegrant strains YSR14 through YSR17 were grown for 8 h in dextrose-containing media to deplete Dad2 expressed under the *GAL* promoter. Cells were harvested and analyzed for cell cycle progression, spindle dynamics, and kinetochore–MT orientation post depletion (Fig 2A). Upon 8 h of growth in repressive media, flow cytometric analysis of propidium iodide-stained cells revealed the parent strain YSR13 (*GALpr-DAD2*) was arrested at the G2/M stage in line with previous observations (Fig 2B). The impaired MT-binding because of the lack of any Dam1 complex subunits is known to activate the spindle assembly checkpoint (SAC) (Hofmann et al, 1998; Janke et al, 2002; Thakur & Sanyal, 2011), resulting in the observed G2/M arrest. This arrest was rescued when the mutant was complemented with ScDad2-FL or ScDad2-R128A as observed in YSR14 and YSR16, respectively, (Figs 2B and S3C). However, no rescue in the arrest was observed in strains YSR15 and YSR17 expressing ScDad2-R126A and ScDad2-ΔDSS, respectively, suggesting the conserved arginine R126 in the DSS motif is essential for mitotic progression in *S. cerevisiae*.

At the same hour of Dad2 protein depletion, we examined the spindle dynamics using the fluorescently tagged spindle pole body protein, Spc110-mCherry, as the marker. In line with the metaphase arrest observed upon depletion of Dad2 in YSR13, we found these cells to have a short mitotic spindle (<2 µm) (Fig 2C). The conditional mutant strain, when complemented with ScDad2-FL as in YSR14 or with ScDad2-R128A as in YSR16, was able to transit metaphase and enter anaphase, as suggested by an increased average spindle length of >2 µm (Fig 2C). We did not observe the rescue of the mitotic arrest and a corresponding increase in the spindle length when the same conditional mutant was complemented with ScDad2-R126A as in YSR15 or ScDad2-ΔDSS as in YSR17 (Fig 2C).

To validate if the observed defects were consequential of improper kinetochore–MT orientation at metaphase, we studied localization of centromeres (marked by *CEN3*-GFP) relative to the SPBs (marked by Spc110-mCherry) to monitor the nature of these attachments (Umbreit et al, 2014). Sister kinetochores were bioriented in most of the cells in YSR14 expressing ScDad2-FL or YSR16 expressing ScDad2-R128A (Fig 2D and E). By contrast, a significant proportion of YSR15 cells expressing ScDad2-R126A or YSR17 cells expressing ScDad2-ΔDSS exhibited monooriented kinetochores as in the case in YSR13 upon depletion of Dad2. Together, these

observations suggest that the conserved arginine residue R126 in the DSS motif plays a critical role in facilitating biorientation of the kinetochores, lacking which, cells remain arrested at metaphase. The observation that the phenotype of mutants expressing ScDad2-R126A or ScDad2-ΔDSS are comparable with the depletion of Dad2 suggests the DSS region, particularly its R126 residue, plays a significant role in the proper functioning of the Dam1 complex in *S. cerevisiae*.

## DSS is critical for oligomerization and MT binding of Dam1 complex

Our genetic studies clearly suggest that the kinetochores bearing mutant versions of Dad2 are unable to attach or remain attached with the MT ends resulting in an increased proportion of cells with monooriented kinetochores. We wondered if the observed phenotype in DSS mutants is indicative of structural changes in the Dam1 complex.

To understand this further, we modified the sequenced encoding for Dad2 in the previously reported polycistronic expression vector (Miranda et al, 2005) encoding all 10 subunits of the ScDam1 complex to express ScDad2-R126A or ScDad2-ΔDSS instead of WT Dad2. We then expressed and purified three versions of the Dam1 complex from *E. coli* containing Dad2-FL (DASH^WT), Dad2-R126A (DASH^R126A), or Dad2-ΔDSS (DASH^ΔDSS), respectively. During the purification process, the DASH^WT eluted as a single peak upon gel filtration (Fig 3A, peak fraction highlighted in gray). On the other hand, both DASH^R126A and DASH^ΔDSS eluted later than the DASH^WT. Furthermore, we also observed a second peak around Ve = 18 ml in the mutants that were more pronounced than that observed in the DASH^WT profile. These are suggestive of structural or/and compositional changes in the complex. We analyzed the copurifying subunits from each of the mutant peak fractions (b for DASH^R126A and d for DASH^ΔDSS) to that of DASH^WT by SDS–PAGE. We also included the fractions marked "a" and "c" in the elution profile of the mutants that corresponded to the peak fraction from the DASH^WT purification. We find that the DASH^R126A complex does not contain the subunit Spc19 indicative of an incomplete complex in the presence of this mutation (Fig 3B). Intriguingly, this was not the case with DASH^ΔDSS which contained a band corresponding to Spc19 despite eluting at a higher volume as compared with DASH^WT. These observations clearly suggest that disruptions in the DSS affect the structure of the complex.

We next tested if the changes described above affected their ability to oligomerize into rings around MTs, a defining feature of the Dam1 complex. The proteins derived from both the peaks for DASH^R126A and DASH^ΔDSS were used in these experiments. In the presence of MTs, we could detect the rings formed by the DASH^WT complex by negative stain electron microscopy (Fig 3C). However, the mutant complexes DASH^R126A and DASH^ΔDSS failed to form such structures indicating that the DSS region is essential for oligomerization of the Dam1 complex to form higher assemblies like the ring (Fig 3C).

Previous studies on the Dam1 complex have identified oligomerization-deficient mutants that still retained the ability to bind MTs. We tested this property in our mutant complexes by MT-cosedimentation assay. The DASH^WT was consistently detected in the pellet fraction in all the conditions we tested (0.5 µM–2 µM MTs in

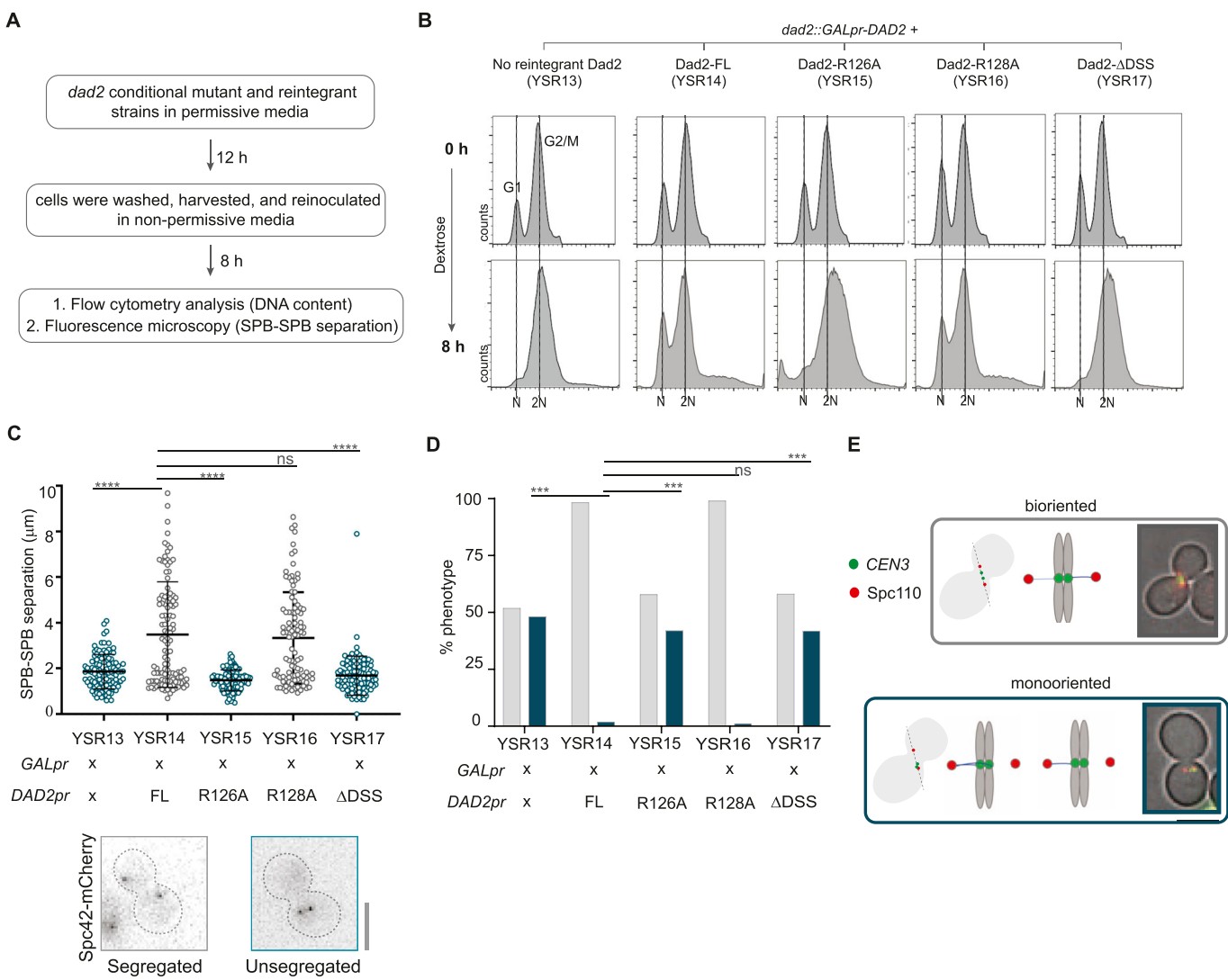

**Figure 2. Alanine substitution of the conserved arginine R126 in Dad2 results in defective spindle dynamics and monooriented kinetochores in *S. cerevisiae*.**
**(A)** Flowchart of the experimental design that was followed to study the contribution of the DSS in Dad2 function in *S. cerevisiae* using the strains YSR13 through YSR17.
**(B)** Histograms indicate the distribution of cells with N (G1 cells) and 2N (G2/M cells) DNA content (*x*-axis) in the *dad2* conditional mutant strain YSR13 and the reintegrant strains YSR14 through YSR17 upon depletion of endogenous Dad2 for 8 h followed by propidium iodide staining. **(C)** The Dad2 conditional mutant strain and the reintegrant strains were analyzed by fluorescence microscopy to study spindle dynamics (using Spc110-mCherry) in each of these strains. The spindle length derived from the SPB–SPB distance (*y*-axis) in each of these strains upon depletion of endogenous Dad2 for 8 h is shown. Only large-budded cells with BI > 0.65 were used for analysis. Statistical significance was tested by one-way ANOVA (****, *P* < 0.0001, n > 100). Representative images of large-budded cells with SPB separation that correspond to WT-like spindle and short spindle are shown below the graph. Scale bar, 5 µm. **(D)** The kinetochore orientation in the indicated strains were studied by localizing *CEN3*-GFP with reference to Spc110-mCherry. The bar plots represent the proportion of cells with bioriented (gray) or monooriented kinetochores (*CEN3*-GFP) (blue) in each strain. Cells with two *CEN3*-GFP puncta between the two SPBs were considered bioriented, and those with only one *CEN3*-GFP punctum closer to one of the two SPBs were considered monooriented. Only large-budded cells with BI > 0.65 were used for analysis. Statistical significance was tested by one-way ANOVA (***, *P* < 0.001, n > 100).
**(E)** Schematic representation of bioriented and monooriented kinetochore–MT attachments is shown along with a representative microscopic image (*right*). Scale bar, 5 µm.
Source data are available for this figure.

twofold increments) suggesting that they physically bound to MTs as reported previously (Figs 3D and S4). In contrast, the binding of both DASH$^{R126A}$ and DASH$^{\Delta DSS}$ to the MTs was severely compromised across the conditions tested. Our results suggest that the DSS domain plays a significant role in both the structure (oligomerization) and function (MT binding) of the Dam1 complex in *S. cerevisiae*. These defects are consistent with our observations in vivo and can be correlated with the increased frequency of monooriented kinetochores in DSS mutants.

It is known that Dad2 is essential for viability in both *C. albicans* and *S. cerevisiae* (Hofmann et al, 1998; Janke et al, 2002; Thakur & Sanyal, 2011). Whereas the DSS mutants were viable in *C. albicans*, they were found lethal *S. cerevisiae*. Considering our observations described above, it is possible that CaDam1 complex is structurally divergent from the budding yeast version, or the kinetochore–MT interface in *C. albicans* shows sufficient plasticity to tolerate variations in the Dam1 complex. To better understand the observation

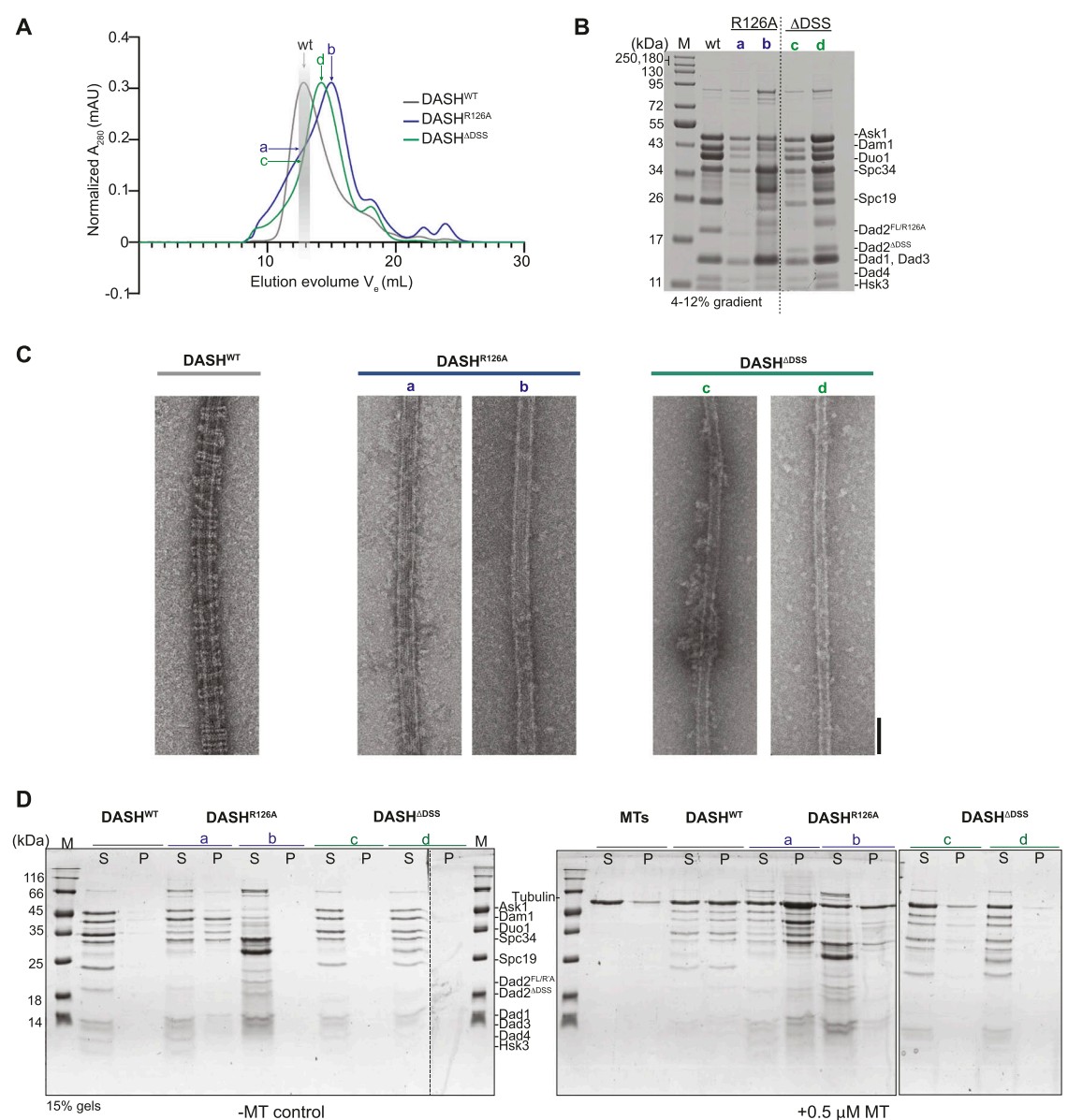

**Figure 3. DSS is critical for oligomerization of Dam1 complex into rings and their association with MTs.**
**(A)** Size exclusion chromatography profile of the WT Dam1 complex and the two mutant versions that contain Dad2-R126A (DASH^R126A) and Dad2-DSS (DASH^ΔDSS), respectively. The elution volume (V_e) corresponding to the peak fraction of DASHWT is marked by a light gray box. The fractions of DASH^R126A and DASH^ΔDSS that match the elution time of DASH^WT are marked a and c, respectively. The peak fractions of the mutants are marked b (for DASH^R126A) and d (for DASH^ΔDSS). **(B)** The WT eluate along with the fractions marked a, b, c, d from the mutant Dam1 complexes were analyzed by SDS–PAGE, using 4–12% gradient gel. M, molecular weight marker. **(C)** Negative stain EM micrographs of MT decoration by DASH^WT, DASH^R126A (from fractions a and b), and DASH^ΔDSS (from fractions c and d) complexes. Scale bar, 50 nm. **(D)** Coomassie stained gels of the co-sedimentation assay to test the ability of indicated versions of Dam1 complex with MTs. Lanes marked S and P correspond to supernatant and pellet fractions from the assay. M, molecular weight marker.

in *C. albicans*, we next studied the kinetochore integrity and chromosome segregation fidelity of these mutants in further detail.

### The conserved arginine residue in the DSS is essential for faithful chromosome segregation in *C. albicans*

To test for functional conservation of the DSS in *C. albicans*, we used the strains ASR02, ASR03, and ASR04 expressing CaDad2-FL, CaDad2-R92A, and CaDad2-ΔDSS, respectively, from the native

genomic locus. A typical feature of *C. albicans* kinetochore is that any defect/mutation compromising the structural integrity of the kinetochore results in the disintegration of the kinetochore ensemble followed by proteasomal degradation of CENPA (Thakur & Sanyal, 2012). We used this property as a test to assess the integrity of the kinetochore upon mutations in the DSS. We engineered the strains ASR02 through ASR04 to express protein-A tagged CENPA. Immunoblotting revealed comparable levels of CENPA between ASR07 cells expressing CaDad2-FL, and the mutant strains ASR08

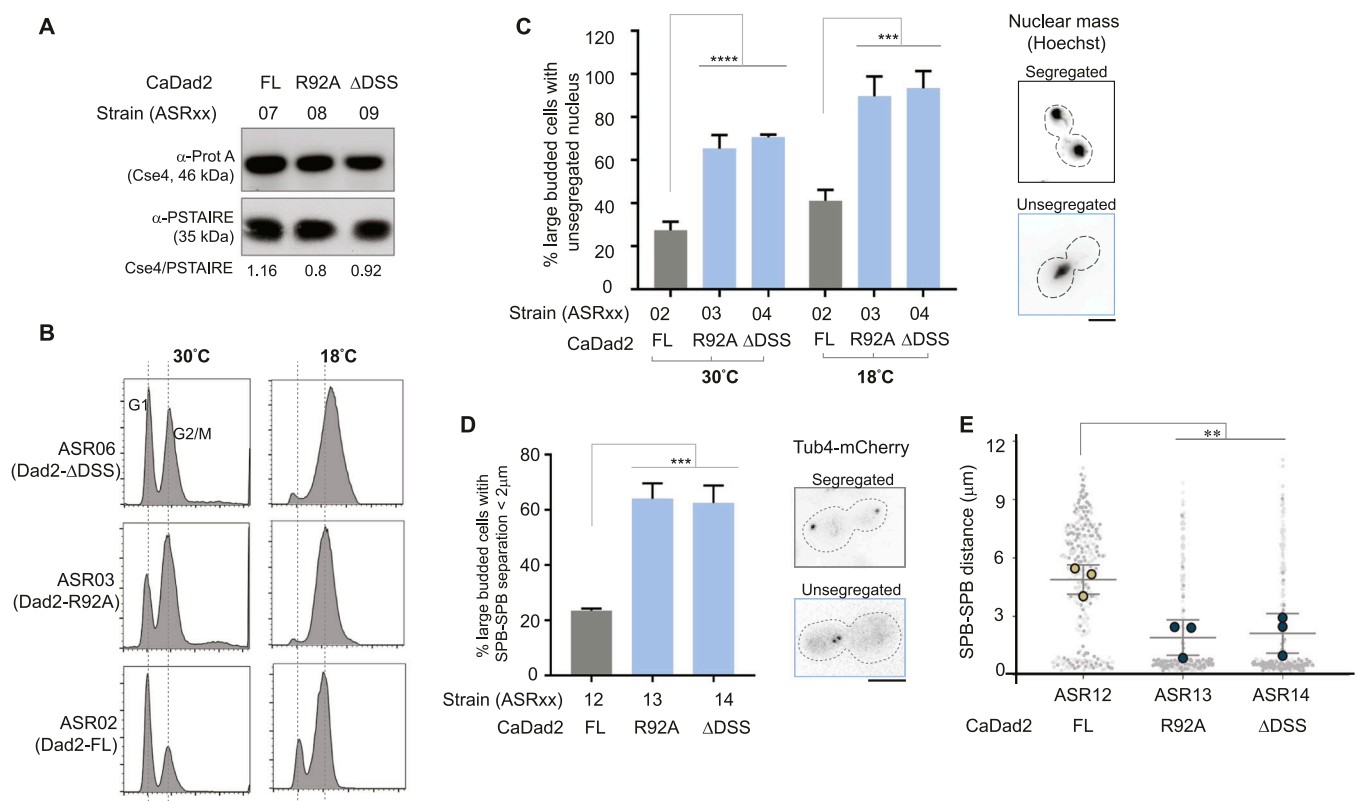

**Figure 4. Alanine substitution of the conserved arginine residue in the DSS is tolerated in *C. albicans*.**
**(A)** The cellular levels of CENPA in strains ASR07 through ASR09 expressing the indicated versions of Dad2 along with a protein-A-tagged CENPA. Whole-cell extracts were prepared from cells grown at 30°C and probed with anti-protein A and anti-PSTAIRE antibodies. The relative CENPA level normalized to PSTAIRE in each sample is indicated below the blot. **(B)** Strains expressing indicated versions of Dad2 were grown in YPDU overnight at 30°C. Cells were reinoculated in YPDU to 0.2 OD$_{600}$ and allowed to grow for two generations at 30°C and 18°C. Histograms depict the distribution of cells with 2N and 4N DNA content (*x*-axis) after flow cytometry analysis. **(C)** The bar graph indicates the proportion of large-budded cells (BI > 0.65) showing unsegregated nuclei in strains ASR02 through ASR04, expressing indicated versions of Dad2, after growth at 30°C and 18°C as shown. Data from three independent experiments were used to generate the plot, and statistical significance was tested by one-way ANOVA (****$P < 0.0001$, ***$P < 0.0003$, n > 100). A representative image of segregated and unsegregated nuclear mass is shown (*right*). The signals are inverted in the image for visualization. Scale bar, 5 *μm*. **(D)** Strains ASR12 through ASR14, expressing indicated versions of Dad2 and Tub4-mCherry, were grown to log phase in YPDU at 30°C and imaged using a fluorescence microscope. The percentage of large-budded cells (BI > 0.65) showing closely placed SPBs (<2 *μm*) in each strain is plotted. Images representative of the two classes of SPB separation observed is shown in the right. Scale bar, 5 *μm*. **(E)** The scatter plot shows the distance between the SPBs in the strains ASR12 through ASR14. The gray circles represent the distribution of data from all three replicates. Circles with a black border represent the mean of each experiment, and the error bars indicate SD among the mean values. Only large-budded cells with BI > 0.65 were included in the analysis. Data from three independent experiments were used to generate the plots. Statistical significance was tested by one-way ANOVA (***$P < 0.0008$, **$P < 0.0095$, n > 100).
Source data are available for this figure.

and ASR09, expressing CaDad2-R92A and CaDad2-ΔDSS, respectively (Fig 4A). The absence of CENPA degradation in this species unlike the case with depletion of Dad2 or other Dam1 complex subunits (Thakur & Sanyal, 2012), is a strong indicator of an intact kinetochore ensemble. This is further supported by the detection of punctate localization signals of Dad2 mutants with similar fluorescence intensities as compared with WT Dad2 (Fig 1J). Together, our observations suggest that neither the deletion of the DSS nor substitution in the conserved arginine residue in the DSS affects the kinetochore assembly in *C. albicans*.

To understand the growth defects of *dad2* mutants observed at a lower temperature, we grew these strains overnight at 30°C, reinoculated them to fresh media in duplicates, and allowed them to complete two generations each at 30°C and 18°C, respectively. Analysis of cell cycle progression of strains grown at 30°C revealed a modest increase in the proportion of cells at the G2/M stage when

they expressed CaDad2-R92A as in ASR03 or CaDad2-ΔDSS as in ASR04 when compared with the control strain ASR02 that expressed CaDad2-FL (Figs 4B and S5A). The frequency of G2/M arrested cells was further amplified when the DSS mutants were grown at 18°C wherein most of the cells displayed this phenotype (Fig 4B). Given that the kinetochore assembly remained unaffected in the DSS mutants (Figs 1J and 4A), we suspected the increase in cells at G2/M and cold-sensitive phenotype of these mutants to be consequential of aberrant kinetochore–MT interactions.

When cells were collected for flow cytometry analysis, additional aliquots of similarly grown cells were examined to analyze their nuclear segregation patterns in the strains ASR02, ASR03, and ASR04 expressing CaDad2-FL, CaDad2-R92A, and CaDad2-ΔDSS, respectively. We observed ~twofold increase in the frequency of unsegregated nuclei in strains expressing CaDad2-R92A and CaDad2-ΔDSS as compared with cells expressing CaDad2-FL when grown at 30°C

(Fig 4C). This frequency further increased to >threefold when these strains were grown at 18°C (Fig 4C). Each of these strains was engineered to express an SPB marker Tub4-mCherry to study the spindle dynamics and correlate them with the frequency of unsegregated nuclei. In the strains ASR13 and ASR14 expressing CaDad2-R92A and CaDad2-ΔDSS, respectively, we observed a significant increase in the frequency of large-budded cells showing short spindles (<2 µm) as compared with ASR12 that expressed CaDad2-FL at 30°C (Fig 4D). This was further reflected in the reduction in the average spindle length in these mutant strains (Fig 4E). We were unable to perform this assay at 18°C as the DSS mutants showed an aberrant elongated large-budded phenotype at this temperature, suggesting that epitope-tagged Tub4 could be nonfunctional in this condition. The increased proportion of cells at metaphase with the spindle length (SPB–SPB distance) of < 2 µm and an unsegregated nuclear mass is suggestive of a delayed anaphase onset in these mutants.

The observed delay in mitotic progression could be because of the inability of these mutants to achieve a bioriented state, as observed in *S. cerevisiae* (Fig 2D). Such defective kinetochore–MT attachments which do not generate tension are detected by sensors like the Aurora B kinase (Ipl1 in yeast) that eventually activate the SAC until proper biorientation is achieved (Biggins & Murray, 2001; Tanaka et al, 2002; Franck et al, 2007; Akiyoshi et al, 2010). We suspected that the accumulation of cells with metaphase-like spindle in the DSS mutants could also be because of a similar defect. As mentioned above, ASR03 (CaDad2-R92A-GFP) and ASR04 (CaDad2-ΔDSS-GFP) cells were arrested completely at the G2/M stage when grown at 18°C. We selected this condition to score for bypass of the G2/M arrest in the corresponding checkpoint-deficient mutants ASR17 (Δmad2 CaDad2-R92A-GFP) and ASR18 (Δmad2 CaDad2-ΔDSS-GFP) by flow cytometry (Fig S5B). The presence of unbudded cells (2N population) in ASR17 and ASR18 mutants as compared with the G2/M-arrested phenotype in ASR03 and ASR04 cells suggested that the DSS mutants were indeed arrested at metaphase. We also analyzed these strains for segregation of the sister kinetochores marked by CaDad2-GFP when grown at 30°C and 18°C (Fig S5C). At both the temperatures tested, the percentage of large-budded cells with unsegregated sister kinetochores in the mother bud (<2 µm between CaDad2 puncta) was reduced in the checkpoint-deficient Dad2 mutants ASR17 and ASR18 as compared with their checkpoint-proficient parent strains, further validating a delay in anaphase onset in these cells.

By comparing the results obtained thus far in *S. cerevisiae* and *C. albicans*, it is evident that the function of the DSS is conserved between these species, with an obvious reduction in the severity of the DSS mutant phenotype in *C. albicans*. The Dam1 complex is known to play a critical role in chromosome segregation by strengthening the kinetochore–MT associations, especially in instances where a single MT binds to a chromosome like in *S. cerevisiae* and *C. albicans*. Although the number of MTs binding a chromosome is the same between these species, a single CENPA nucleosome forms the basis of MT attachment in *S. cerevisiae* (Winey et al, 1995). On the other hand, one among four CENPA nucleosomes anchors the MTs in *C. albicans* (Joglekar et al, 2008). We speculated that the presence of additional CENPA nucleosomes confers a degree of tolerance towards mutations in the DSS. To test

if this correlation indeed exists, we studied the functional significance of the DSS in the ascomycete *S. pombe*, which is a classic example for the large regional centromere structure.

## DSS is dispensable for Dad2 function in *S. pombe*

Unlike any of the species we tested earlier, *dad2* mutants are viable in *S. pombe* (Liu et al, 2005; Kobayashi et al, 2007). However, these mutants undergo prolonged mitosis because of biorientation defects that result in lagging chromosomes. At lower temperatures, these mutants missegregated chromosomes causing a cold-sensitive phenotype (Kobayashi et al, 2007). We exploited this property to study functional significance of DSS in SpDad2 as follows. Null mutants of *dad2* in a strain where the chromosomes are tagged by *lacO*-LacI-GFP system have been reported previously (Strain 1619, [Blyth et al, 2018]). We independently reintegrated SpDad2-FL (PSR01) and SpDad2-ΔDSS (PSR02) into the native genomic locus with their expression driven by the native Dad2 promoter (Fig 5A).

Spot dilution assays did not reveal any apparent growth difference between *dad2* null mutants and the reintegrant strains PSR01 and PSR02 expressing SpDad2-FL and SpDad2-ΔDSS, respectively, at ambient growth temperatures (Fig 5B). Reintegrating Sp-Dad2FL rescued the cold-sensitive phenotype observed in *dad2* null mutant strain SP1619 as expected. Furthermore, we also observed a rescue in PSR02 cells expressing SpDad2-ΔDSS, suggesting that DSS is not essential for Dad2 function in *S. pombe*. We strengthened our correlation further by testing for the essentiality of DSS in Dad2 function in another large regional centromere containing species *C. neoformans*. Unlike the species studied above, *C. neoformans* belongs to Basidiomycota and diverged from a common ancestor it shared with *S. pombe* ~642 MYA (Kumar et al, 2017).

## The conserved arginine R102 in CnDad2 is dispensable for mitotic progression in *C. neoformans*

We generated constructs to introduce an alanine substitution mutation in the conserved arginine residue (R102 in CnDad2) and for the deletion of the entire DSS region in CnDad2 and express them along with mCherry tagged at the C-terminus in *C. neoformans* (Fig 5C). As a control, another construct to express an unaltered full-length CnDad2 sequence was also generated. Each of these constructs was used to transform the strain CNV108 (expressing GFP-H4) (Kozubowski et al, 2013) to generate CNSD169 (CnDad2-FL-mCherry, GFP-H4), CNSD170 (CnDad2-R102A-mCherry, GFP-H4), and CNSD171 (CnDad2-ΔDSS-mCherry, GFP-H4).

Spot dilution assays performed at routine growth conditions in YPD at 30°C revealed that neither the point mutation R102A nor the deletion of the entire DSS motif of CnDad2 resulted in an obvious growth defect, suggesting that the DSS mutants are viable in *C. neoformans* as well (Fig 5D). We also examined growth inhibitions of these mutants at a lower temperature (14°C) or in the presence of thiabendazole (6 µg/ml), conditions used to test mutants for defective kinetochore–MT associations. We failed to observe any growth defects in the mutant strains CNSD170 and CNSD171 expressing CnDad2-R102A and CnDad2-ΔDSS, respectively, as

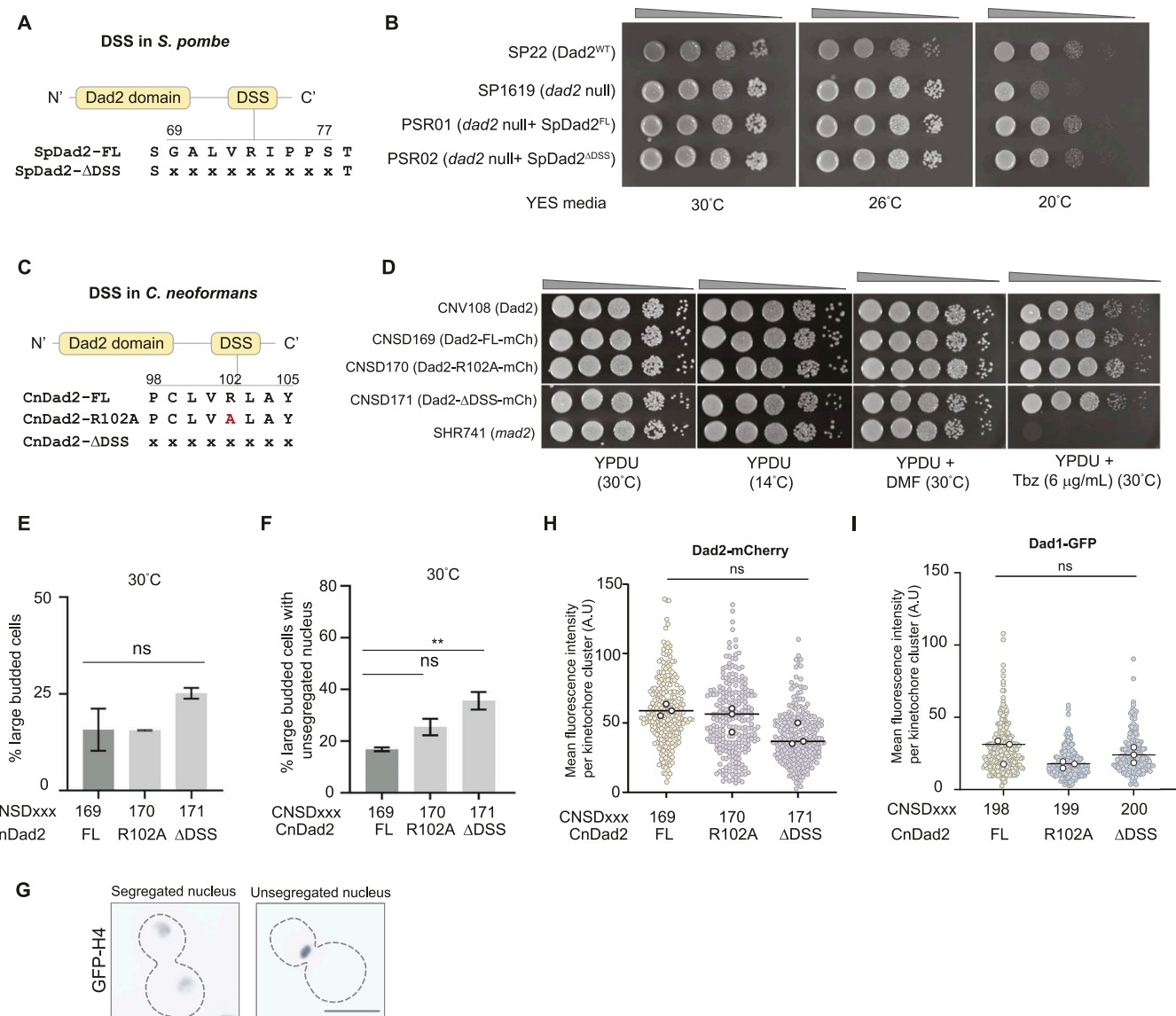

**Figure 5. The DSS is dispensable for viability in species with large regional centromeres.**
**(A)** A schematic representation of the DSS sequence and is position in Dad2 and the residues that were deleted to make the construct SpDad2-ΔDSS. **(B)** Spot dilution assay using the indicated strains grown in YES media grown at 26°C for 24 h. Cells were serially diluted 10-fold and spotted ($10^5$ to $10^2$ cells). Plates were photographed after incubation for 3 d at 30°C and 26°C, and for 5 d at 26°C. **(C)** Schematic depicts the position and sequence of amino acids in the DSS in *C. neoformans* Dad2 and the mutants generated (CnDad2-R102A and CnDad2-ΔDSS) to study the function of the conserved arginine in the DSS in this species. **(D)** Spot dilution assay performed with the indicated strains grown in YPDU for 14 h at 30°C. Cells were serially diluted 10-fold ($10^5$ to 10 cells) and spotted on YPDU plates with the indicated additives and incubated at 30°C. One YPDU plate was incubated at 14°C to test for cold sensitivity. Plates at 30°C were imaged 48 h post-incubation. The plate incubated at 14°C was imaged after 5 d. **(E)** The proportion of large-budded cells in strains CNSD169 through CNSD171 expressing CnDad2-FL, CnDad2-R102A, and CnDad2-ΔDSS after logarithmic growth at 30°C in YPDU as observed under a microscope (n > 100). **(F)** The percentage of large-budded cells with an unsegregated nuclear mass is plotted for strains CNSD169 through CNSD171 (n > 100, unpaired *t* test with Welch's correction, **, *P* < 0.01). **(D, G)** Images are representative of properly segregated and unsegregated nuclear mass, the two phenotypes analyzed in (D). Scale bar, 5 *μ*m. **(H)** The mean fluorescence intensity of the indicated versions of Dad2 is plotted. The scatter plot for each strain represents the distribution of measurements from all the replicates. The black circles represent the mean value from each replicate. No statistical significance was detected (*t* test with Welsch's correction). **(I)** The mean fluorescence intensity of Dad1 in strains expressing the indicated versions of Dad2 was plotted. The scatter plot for each strain represents the distribution of measurements from all the replicates. The black circles represent the mean value from each replicate. No statistical significance was detected (*t* test with Welsch's correction).

compared with CNSD169 cells expressing CnDad2-FL (Fig 5D). To corroborate these observations, we grew each of these strains to log phase at 30°C. We did not observe a significant increase in the number of large-budded cells in strains expressing WT or mutant versions of Dad2-bearing alanine substituted conserved arginine residue in the DSS or deletion of the entire DSS itself (Fig 5E). However, further analysis of nuclear segregation by localization of GFP-tagged histone H4 revealed that the CNSD171 mutant strain

expressing the DSS deletion, but not CNSD170 with the point mutation CnDad2-R102A, had a significant increase in the proportion of large-budded cells with an unsegregated nuclear mass (Fig 5F and G). We do not find any significant difference in the fluorescence intensities between CnDad2-FL, CnDad2-R102A, and CnDad2-ΔDSS, suggesting that the kinetochore structure was not compromised by the mutations introduced (Fig 5H). We validated this further by testing the levels of another outer kinetochore protein Dad1in the strains CNSD172, CNSD173, and CNSD174 expressing CnDad2-FL, CnDad2-R102A, and CnDad2-ΔDSS, respectively (Fig 5I).

These results suggest that the conserved arginine residue in Dad2 is dispensable for viability and timely mitotic progression in *C. neoformans* unlike in *S. cerevisiae* or *C. albicans*. However, a small but significant increase in the large bud arrested cells when the entire DSS was deleted, suggesting the importance of the motif in chromosome segregation in *C. neoformans*.

The tolerance of *S. pombe* and *C. neoformans* to mutations in the DSS is suggestive of a nonessential role for the conserved arginine residue/DSS in Dam1 complex function in these species as compared with *S. cerevisiae* or *C. albicans*. Despite a high level of conservation at the amino acid sequence in the DSS, the striking differences in the dependence on this conserved arginine residue for mitotic progression across the evolutionarily diverged species represent a strong inverse correlations with the centromere length.

## Discussion

In this study, we identify an evolutionarily conserved domain, the DSS, at the C-terminus of Dad2 that hosts a highly conserved arginine residue. Our phylogenetic analysis of sequenced genomes across Ascomycota, Basidiomycota, and some species of Mucoromycota, reveals that the arginine residue remains conserved for more than 700 million years. Alanine substitution analysis suggests that this arginine residue in the evolutionarily conserved DSS motif is essential for viability and timely mitotic progression in *S. cerevisiae* as the mutant cells irreversibly arrested at the G2/M stage with a metaphase-like spindle. The presence of monooriented kinetochores in the DSS mutants in *S. cerevisiae* highlights the functional significance of the conserved arginine residue in establishing sister chromatid biorientation. This could be correlated with the inability of mutant Dam1 complex to form rings and bind to MTs in vitro. We observed similar mitotic defects, albeit at a lower frequency, when the corresponding arginine residue was mutated to alanine in Dad2 in *C. albicans* harboring small regional centromeres. A small but significant increase in mitotic defects was observed only upon deletion of the entire DSS in *C. neoformans* that possesses large regional centromeres. The alanine substitution of the conserved arginine in the DSS is neither lethal nor resulted in accumulation of cells with unsegregated nuclei, suggestive of a redundant function conferred by the arginine residue in *C. neoformans*. In *S. pombe*, another representative of the species with large regional centromeres, DSS was found dispensable in rescuing the cold-sensitive phenotype of *dad2* mutants suggesting a functional redundancy. The diminishing significance of the DSS with increasing lengths of centromeric chromatin (Fig 6A) tempted us to

speculate that larger centromeres may have evolved to develop tolerance to mutations that cause lethality to an organism possessing point centromeres.

The Dam1 complex is responsible for the establishment and maintenance of sister chromatid biorientation until the SAC is satisfied (Cheeseman et al, 2001a; Janke et al, 2002; Franco et al, 2007; Tanaka et al, 2007). Conditional mutants of Dam1 complex subunits arrest at the large-budded stage with short spindles and an unsegregated nuclear mass (Cheeseman et al, 2001b; Janke et al, 2002; Li et al, 2002). We find a similar phenotype upon alanine substitution of the conserved arginine in the DSS in *S. cerevisiae*. This suggests that the DSS, particularly the conserved arginine R126, could play an essential role in maintaining the overall integrity of the Dam1 complex in vivo. Some of the known factors that govern the function of the Dam1 complex include (a) its interaction with the MTs and the Ndc80 complex for kinetochore localization (Janke et al, 2002; Li et al, 2002; Wong et al, 2007; Maure et al, 2011; Lampert et al, 2013; Kim et al, 2017), and (b) its ability to oligomerize to a ring-like structure around MTs that enables it to sustain its MT association under tension (Miranda et al, 2007; Umbreit et al, 2014; Zelter et al, 2015). Our biochemical reconstitutions indeed find disruption of both these aspects of the Dam1 complex function by R126A mutation.

In the known structure of the Dam1 protomer (Jenni & Harrison, 2018), we mapped the C terminus of Dad2 to the central core domain known to be essential for the stability of the Dam1 monomer. The amino acids in the DSS were found in interacting proximity with the Spc19, Spc34, and Dam1 subunits (Fig S6). Consistently, Spc19 failed to coelute with other subunits when Dad2FL was replaced with Dad2R126A in the DASH expression vector. The resulting mutant complex lacking Spc19 was unable to oligomerize into rings even in the presence of MTs. To understand this further, we analyzed the interface between Dam1 monomers from previous crosslinking analysis and coarse grain models (Zelter et al, 2015; Legal et al, 2016). Contacts between Spc34 and Ask1 of two Dam1 monomers form the major interaction interface between them. This is further supported by the contacts between Duo1 from one monomer and Spc19 and Ask1 of another. The loss of Spc19 subunit like the case in R126A mutant could have a direct impact on oligomerization of the complex because of the loss of its interaction with Duo1. The loss of Spc19 can also have a significant effect on the position and interactions made by Spc34, a key player in inter–Dam1 interface, thereby having an indirect impact on oligomerization.

We suspect the observed loss of MT-binding to be a consequence of oligomerization defects. It is known that oligomerization is critical for stable association with MTs under tension (Umbreit et al, 2014). Deletion of Hsk3 subunit resulted in an oligomerization defective 6-mer complex that failed to form rings around MTs and showed a weaker ability to sustain MT binding under tension (Miranda et al, 2007; Umbreit et al, 2014). In the oligomerization-deficient complexes that we purified with mutations in the DSS, we observed very weak/no binding to MTs in the cosedimentation assay across the conditions we tested. Perhaps sensitive techniques like the rupture force assay used by Umbreit et al (2014) would detect residual binding and the loss in binding efficiency. Strikingly, the mutant complexes described in this study and in Umbreit et al (2014) retain the subunits Dam1 and Duo1, which are the only two proteins known to crosslink microtubules within this

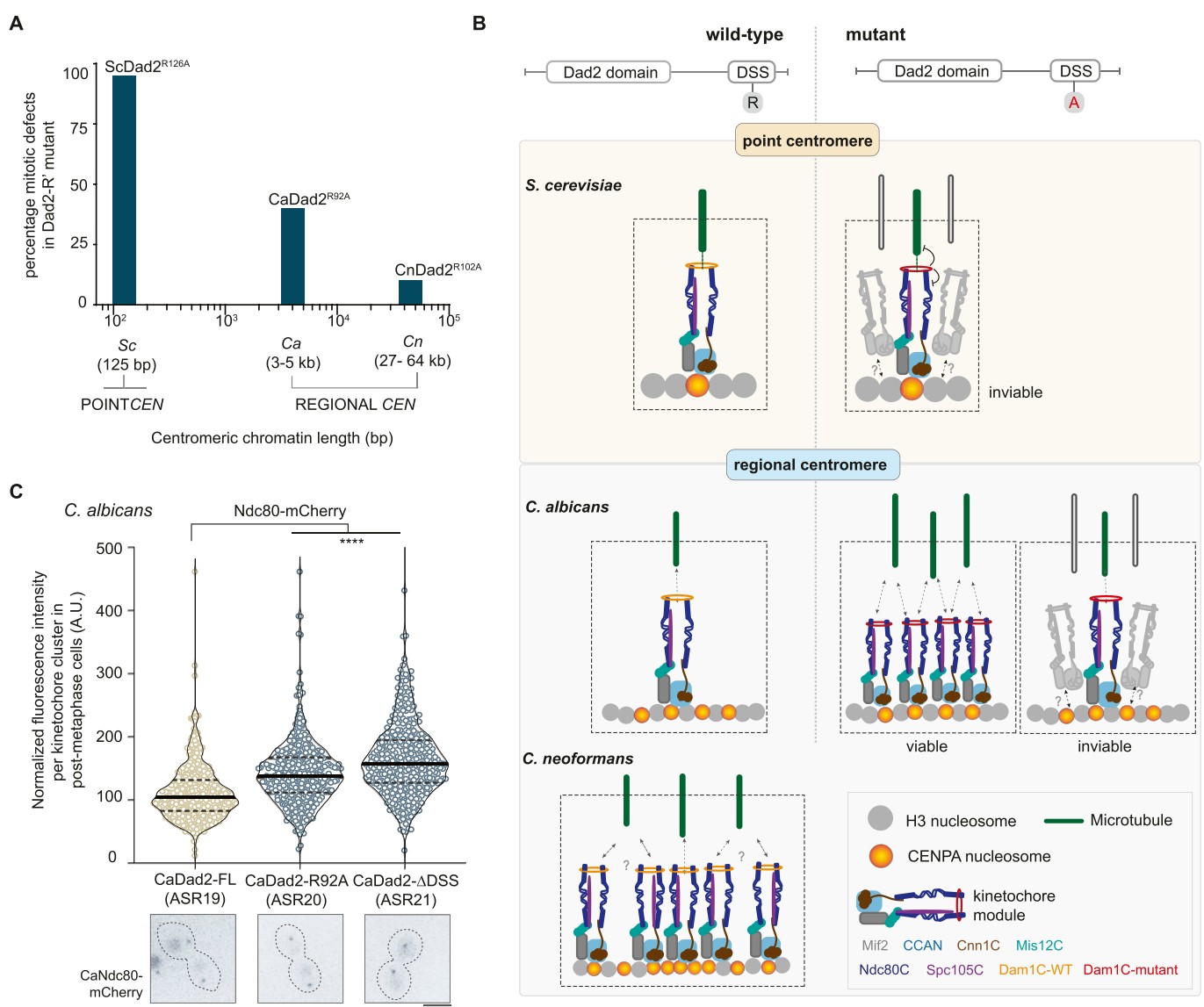

**Figure 6. The tolerance towards alanine substitution of the conserved arginine in the DSS is dependent on the length of centromeric chromatin.**
**(A)** Summary of the alanine substitution mutant phenotype across the three fungal species tested in this study. The length of centromeric chromatin of each species is mentioned in the *x*-axis. The diminishing severity of the mutant phenotype in each of the species studied is represented in the *y*-axis as the frequency of cells showing defective segregation in the form of an unsegregated nuclear mass and/or a short mitotic spindle. Abbreviations in the *x*-axis indicate the following: Sc, *S. cerevisiae*; Ca, *C. albicans*; and Cn, *C. neoformans*. **(B)** A model to explain the effect of mutations in the DSS in organisms with point centromeres versus those with regional centromeres. The kinetochore formed on a single CENPA nucleosome binds to an MT during chromosome segregation in organisms with point centromeres. Mutations compromising the efficiency of this interaction can result in viability loss. In the case of regional centromeres, represented by *C. albicans*, the centromeres possess additional CENPA nucleosomes apart from the one that binds to the MT (Joglekar et al, 2008; Burrack et al, 2011; Thakur & Sanyal, 2011). Cells that can recruit excess outer kinetochore proteins often enable additional kMTs to bind to a chromosome facilitating chromosome segregation even with a mutant kinetochore. The subset of cells unable to recruit excess kinetochore proteins/kMTs eventually loses viability. The inherent property of *C. neoformans* centromeres to bind multiple microtubules as predicted from the size of the centromeric chromatin is shown. This configuration makes the conserved arginine residue in the DSS dispensable for viability in *C. neoformans*. **(C)** *C. albicans* strains ASR19 through ASR21, expressing the indicated versions of Dad2 and Ndc80-mCherry, were grown to log phase at 30°C, washed, and observed under a fluorescence microscope. The normalized fluorescence intensity of Ndc80-mCherry per kinetochore cluster in each strain (*x*-axis) is plotted (*y*-axis). Data from three experiments covering > 350 kinetochore clusters were used to generate the violin plot. Black lines indicate mean and dotted lines mark the quartiles. Statistical significance was tested by one-way ANOVA (****, *P* < 0.0001). Below each strain in the *x*-axis, a representative image for Ndc80 localization in each of them is shown. Scale bar, 5 µm.

complex (Umbreit et al, 2014; Zelter et al, 2015; Legal et al, 2016). Loss of MT binding despite their presence further reiterates the importance of oligomerization on the Dam1 complex function. Consistent with these findings, we do observe biorientation defects in

the DSS mutants that are a mark of weak or unstable kinetochore–MT interactions.

Given that a coupler like the Dam1 complex is indispensable for viability in species with a single kMT binding to a chromosome as in

*S. cerevisiae* and *C. albicans* (Winey et al, 1995; Cheeseman et al, 2001b; Janke et al, 2002; Li et al, 2002; Burrack et al, 2011; Thakur & Sanyal, 2011), one would expect similar consequences to mutations in the DSS in both these species. Whereas the DSS function is conserved in these species, the mutants are viable in *C. albicans*, unlike *S. cerevisiae*. We ascribe the tolerance observed in *C. albicans* to the presence an accessory CENPA cloud at the centromere cluster (Joglekar et al, 2008; Thakur & Sanyal, 2013; Fukagawa & Earnshaw, 2014).

In an organism like *S. cerevisiae*, a single kinetochore module, assembled on 1–2 CENPA nucleosomes, serves as a MT receptor on each chromosome (Joglekar et al, 2006; Furuyama & Biggins, 2007; Cieslinski et al, 2021 *Preprint*). The strength of this association solely determines sister chromatid biorientation and its subsequent segregation. Thus, it is conceivable that the mutations compromising the kinetochore–MT association would not be tolerated in *S. cerevisiae* (Fig 6B). Indeed, several point mutants of subunits of the Dam1 complex have been shown to be lethal (Cheeseman et al, 2001a; Janke et al, 2002). Although *C. albicans* centromeres tether a single kMT, there are ~4 CENPA nucleosomes on each of the 3–5 kb long *C. albicans* centromeres (Joglekar et al, 2008). We speculated that additional kinetochore subunits are recruited to these "spare" CENPA nucleosomes in *C. albicans*, serving as receptors that facilitate the binding of multiple MTs to a chromosome when a "need" arises. Such a failsafe mechanism can ensure proper chromosome segregation even with a weaker mutant kinetochore, like the DSS mutants (Fig 6B). Indeed, we observed significantly higher levels of Ndc80 at the kinetochore cluster in the DSS mutants that were able to segregate their chromosomes in post-metaphase cells as compared with the control WT cells (Fig 6C). The subset of cells that were unable to recruit excess outer kinetochore subunits may fail to segregate chromosomes, thereby losing viability. This clearly demonstrates that the survivors in mutant strains expressing mutant versions of Dad2 such as CaDad2-R92A and CaDad2-ΔDSS are primed to recruit multiple MTs to a chromosome.

Operating such a mechanism requires the availability of surplus kinetochore subunits and the ability to seed the required number of additional MTs, neither of which are limiting in a cell. Overexpression of CENPA in *C. albicans* has been shown to increase the copy number of the outer kinetochore proteins without crossing the WT boundaries of the centromeric chromatin (Burrack et al, 2011). Furthermore, engineered strains in both *C. albicans* and *S. cerevisiae* formed additional kMTs suggesting that the stoichiometry of the number of kMTs to the number of chromosomes can be dynamic (Burrack et al, 2011; Nannas et al, 2014). The lack of such a mechanism in *S. cerevisiae* despite these abilities can be attributed to their point centromere structure. The size restriction and dependence on DNA sequence prevent the enrichment or spreading of kinetochore proteins at the native centromere, thereby impeding the formation of additional MT attachments.

We also propose an alternate, non-mutually exclusive possibility wherein the enrichment of Ndc80 by itself can facilitate efficient MT capture by a weakened kinetochore. Several studies have shown that Ndc80 can form crosslinks with adjacent Ndc80 molecules through the unstructured loop region (Maure et al, 2011; Polley et al, 2023). The consequence of such homotypic crosslinking is a cooperative increase in the affinity to MTs binding and the ability to stay associated with MTs under tension. This translates to the increased stability of end-on attachments in vivo (Hsu & Toda, 2011;

Tang et al, 2013; Helgeson et al, 2018; Polley et al, 2023). Despite the unstructured loop being conserved across eukaryotes, the loop-mediated multivalency of Ndc80 molecules can have a stronger impact on MT binding in large regional centromeres as compared with point centromeres because of the possible limits of Ndc80 enrichment that can be accommodated.

When compared with *C. albicans*, we find mutations in the DSS to be tolerated better in *S. pombe* and *C. neoformans* in physiological and stress conditions like low-temperature growth or in the presence of sublethal doses of spindle poisons like thiabendazole. Although we did not directly score for the mitotic progression or chromosome segregation fidelity in *S. pombe*, we observed the rescue of cold-sensitive phenotype of *dad2* mutants that resulted from defective mitosis, even with SpDad2-ΔDSS. Unlike *S. pombe*, depleting Dad2 results in viability loss in *C. neoformans* (Sridhar et al, 2021) which could be rescued with CnDad2-ΔDSS. Both these species harbor large regional centromeres. It is known that the *S. pombe* chromosomes have ~10-kb-long CENPA-enriched centromeric chromatin with 10–15 CENPA nucleosomes that collectively tether 2–3 kMTs (Ding et al, 1993; Joglekar et al, 2008; Coffman et al, 2011; Rhind et al, 2011). Based on these estimates, the *C. neoformans* centromeres that span 27–64 kb are expected to have at least 10 CENPA nucleosomes that collectively tether multiple kMTs (>2 kMTs) per chromosome. This is further supported by the fact that ~4 kMTs bind each chromosome in chicken and flies wherein the centromeres are longer than 30 kb (Fig S7A) (Ribeiro et al, 2009, 2010; Shang et al, 2010). The dispensability of DSS in two phylogenetically distantly related species with longer centromeric chromatin reinforces the proposed inverse correlation with centromere length.

Large centromeres are known to act as a buffer-enabling harmless kinetochore drift across the centromere to prevent functional interference of neighboring essential genes (Fukagawa & Earnshaw, 2014). In this study, we uncover another potential advantage associated with regional centromeres in tolerating conditions suboptimal for chromosome segregation. This adaptive edge provided by regional centromeres in enabling compensatory mechanisms may explain their recurring presence across Eukaryota (Fig S7B and Table S1). Our results also highlight a significant role of centromere size/length in shaping the evolution of Dam1 complex function and its physiological structure. This can be attributed to several features such as the presence of excess CENPA nucleosomes, tethering multiple kMTs per chromosome, and/or enabling multivalency of Ndc80. The ring structure could be an evolutionarily selected conformation that is indispensable for the attachment of chromosomes with point centromeres, a derived state that provides a single attachment point to the MTs. Further investigations on the Dam1 complex in different fungi will shed light on other functional assemblies beyond the Dam1 rings and their significance.

# Materials and Methods

### Media and growth conditions

*S. cerevisiae*, *C. albicans*, and *C. neoformans* strains used in this study were grown in YPD (2% dextrose, 2% peptone, 1% yeast extract

supplemented with 0.01% adenine) and incubated at 30°C at 180 rpm. Conditional expression strains in *S. cerevisiae* were propagated in galactose media (YPG, 2% galactose, 0.3% raffinose, 2% peptone, 1% yeast extract) unless repression in YPD is mentioned. Transformation of *S. cerevisiae* and *C. albicans* were performed by standard lithium acetate–PEG method (Sanyal & Carbon, 2002; Geitz & Schiestl, 2007). Biolistic transformation method was used for *C. neoformans* strains (Davidson et al, 2000). Selection of transformants was based on prototrophy for the metabolic markers used. In case of antibiotic marker NAT, selection was done in media supplemented with 100 μg/ml nourseothricin (ClonNAT; CAS 96736-11-7; Werner Bioagents). For cold sensitivity assays, the growth temperature was reduced to 18°C for *C. albicans* and 14°C for *C. neoformans*. For thiabendazole sensitivity assay, the growth media were supplemented with indicated concentration of thiabendazole (10 mg/ml stock in dimethyl formamide; Sigma-Aldrich).

## Strains and plasmids

The list of all the strains, plasmids, and primers used are provided in Tables S2–S4 respectively. The construction of each of them is detailed below.

### Construction of *S. cerevisiae* strains

#### Construction of vectors to express Dad2-GFP (WT, mutated or truncated form) in pRS313

To express GFP-tagged ScDad2 from pRS313 (CEN/ARS/HIS1 plasmid), sequences coding for GFP along with terminator sequences of Sc*ACT1* was amplified using primer pair ScGFP-F/ScGFP-R and cloned as a BamHI-ClaI fragment into pRS313 resulting in pRS313G. This plasmid was subsequently used to clone Dad2 with/without mutations. For cloning ScDad2-FL, a fragment containing Sc*DAD2*pr-*DAD2* ORF was amplified from the genomic DNA of BY4741 strain with primers SR282/SR290 and cloned in frame with GFP as a SacII-BamHI fragment into pRS313G. The resulting plasmid was named pSR01. Because the DSS in ScDad2 is located near the stop codon, any desired mutation was incorporated in the reverse primer used to amplify Dad2. In this way, ScDad2-R126A, ScDad2-R128A, and ScDad2ΔDSS were amplified with primer pairs SR289/Sc126A-R, SR289/Sc128A-R, and SR289/ScΔDSS-R, respectively, and cloned into the SacII-BamHI sites of pRS313G. The resulting plasmids were named pSR02 through pSR04. Each of these constructs was confirmed by Sanger sequencing using SR282.

#### Construction of strains YSR01 through YSR05

A previously reported conditional mutant of ScDad2 (CJY077: *MATa Δdad2::KanMX6 ura3-52 lys2-801 ade2-101 trp1Δ63 leu2Δ1::pCJ055[dad2^ts, LEU2] his3Δ200*) was used as the parent strain. The plasmids pRS313G, pSR01–pSR04 were used to transform the CJY077 strain by the standard lithium acetate method to generate strains YSR01 through YSR05, respectively. Transformants were selected on CM-leu-his media upon incubation at 28°C. Selected colonies from the transformation plate were streaked again on CM-leu-his media for single colonies and used for subsequent experiments.

### Construction of strains YSR06 through YSR12

The strain BY4741 was engineered to express Spc42 with mCherry tag at the C-terminus resulting in YSR06. The cassette was amplified from the plasmid pAW8-mCherry using the primer pair S42F/S42R. A Dad2 protection plasmid was constructed as follows. The Sc*DAD2* gene with its promoter and terminator was amplified using the primer pair ScDad2Pr F/ScDad2R and cloned as a SacII/SacI fragment into pRS316 (CEN/ARS/URA) resulting in the plasmid pSR05. The strain YSR06 was transformed with the plasmid pSR05 by the standard lithium acetate method to generate strain YSR07. The transformants were selected on CM-ura media upon incubation at 30°C.

YSR07 cells grown on CM-URA media were then transformed with a cassette to delete *DAD2* ORF with a *LEU* marker. The cassette was amplified from the plasmid pUG73 using the primer pair ScDad2delF/ScDad2delR. Because the deletion cassette can delete *DAD2* from either the native locus or from the plasmid, the desired transformants where the deletion occurred on the genomic *DAD2* locus were selected by their inability to grow on media containing 5'FOA. This strain was named YSR08 and was further confirmed by PCR using the primer pair ScDad2F/Leu2R. Strains YSR09–YSR12 were subsequently generated by independently transforming YSR08 cells with plasmids pSR01–pSR04. The transformants were selected on CM-ura-his media upon incubation at 30°C. The strains YSR09 through YSR12 were subsequently maintained in these media unless stated otherwise.

### Construction of strains YSR13 through YSR17

A cassette to replace the native *DAD2* promoter with the $GAL_{1-10}$ promoter was amplified with the primer pairs GalDad2FP/GalDad2RP using pYM-N25 as the template. The PCR product was used to transform the strain SBY12503 to obtain the conditional mutant strain YSR13. The transformants were selected on YPDU plates supplemented with 100 μg/ml norseothricin upon incubation at 30°C.

For integrating ScDad2-FL, the ORF-containing fragment was amplified using the primers Dad2pr-SacII-F/Dad2pr-SacI-R. A re-integration cassette was generated by cloning the *DAD2* ORF along with the promoter in SacII/SacI sites into pUG73 resulting in pSR06. To introduce mutations corresponding to ScDad2-R126A, ScDad2-R128A, and ScDad2-ΔDSS, the ORF-containing fragments were constructed by overlap PCR (see list of primers). Each of these fragments were independently cloned as SacII-SacI fragment into pUG73 resulting in plasmids pSR07 (ScDad2-R126A in pUG73), pSR08 (ScDad2-R128A in pUG73), and pSR09 (ScDad2-ΔDSS in pUG73).

Each of these plasmids pSR06–pSR09 was linearized by digestion with SpeI and then used to transform the conditional mutant strain YSR13 to obtain reintegrants YSR14–YSR17 that, respectively, express ScDad2-FL, ScDad2-R126A, ScDad2-R128A, and ScDad2-ΔDSS from the DAD2 promoter from the native *DAD2* locus. The transformants were selected on CM(Gal)-leu media upon incubation at 30°C.

### Construction of *C. albicans* strains

#### Construction of strains J108A and J108B

A plasmid pBS-RN was constructed by amplifying the *RPS1* locus using the primer pair RP10F/RP10R and cloned as a NotI fragment

into the pBS-NAT plasmid. A previously reported plasmid pDad2-TAP was used as a template to amplify WT full-length *DAD2* along with its promoter and a C-terminal TAP tag using the primer pair AD02/AD03. The amplicon was cloned as a SalI fragment into pBS-RN resulting in the plasmid pRN-Dad2-FL. The DSS-deleted version was generated by overlap PCR strategy using the primer pairs AD02/Dad2delR and Dad2delF/AD03. The overlap product was cloned as a SalI fragment into pBS-RN to obtain the plasmid pRN-Dad2-ΔDSS. Both pRN-Dad2-FL and pRN-Dad2-ΔDSS were used to transform the conditional *dad2* mutant J108 after linearization with StuI. The transformants were selected on YP-Succinate media supplemented with 200 μg/ml NAT.

### Construction of strains ASR01 through ASR04

A previously reported plasmid pDad2Δ3 (*CaDAD2* deletion with a *HIS1* marker [Thakur & Sanyal, 2011]) was used to transform SN148 after digestion with SacI-KpnI. The resulting heterozygous mutant of *DAD2* was named ASR01. The vectors to incorporate desired mutations in the remaining *DAD2* allele were constructed as follows.

Sequences 537 bp downstream of the *DAD2* ORF was amplified using the primer pair Dad2DS-F/Dad2DS-R and cloned as a XhoI/KpnI fragment into pBS-GFP-Ura plasmid. The resulting plasmid was named pSR10 and was used to clone the other homology region containing *DAD2* ORF with desired mutations. In this regard, mutant versions of *DAD2* were generated by overlap PCR method as described earlier and the product was cloned into pBS-RN resulting in plasmids pRN-92(for CaDad2-R92A). Along with pRN-Dad2-FL and pRN-Dad2-ΔDSS, these plasmids served as intermediate vectors from which *CaDAD2* ORF with the corresponding mutation was amplified using the primer pair Dad2FP/Dad2GFP-RP. Each of these fragments were cloned as a SacII/SpeI fragment into pSR10 resulting in plasmids pSR11 (CaDad2-FL), pSR12 (CaDad2-R92A), and pSR15 (CaDad2-ΔDSS). These plasmids were then used to transform ASR01 after digestion with SacII-KpnI to result in strains ASR02 through ASR04. The transformants were selected in CM-uri media. The presence of desired mutations was confirmed by Sanger sequencing using the primer Dad2cFP.

### Construction of strains ASR07 through ASR09

To tag CENPA with a TAP tag, a previously reported plasmid pCse4TAP-Leu (Varshney & Sanyal, 2019) was linearized with XhoI and used the transform strains ASR02 through ASR04 to obtain the strains ASR07 through ASR09. The transformants were selected on CM-leu plates. Expression of the fusion protein was confirmed by immunoblotting.

### Construction of strains ASR12 through ASR14

A plasmid to tag Tub4 with mCherry was generated as follows. The 3′ part of the *TUB4* gene was released from a previously reported plasmid pTub4GFP-Ura as a SacII/SpeI fragment and cloned into pDam1-mCherryNAT at these sites, replacing *DAM1* with *TUB4*. The resulting plasmid pTub4-mCherryNAT was linearized by XbaI digestion and used to transform the strains ASR02-04 to obtain ASR12 through ASR14. The transformants were selected on YPD plates supplemented with 100 μg/ml NAT. Transformants were confirmed by fluorescence microscopy.

### Construction of ASR17 and ASR18

One allele of *MAD2* in ASR03 (CaDad2-R92A) and ASR04 (CaDad2-ΔDSS) was deleted by transformation with the plasmid pMad2-2 (Thakur & Sanyal, 2011) after digestion by SacII-XhoI. The resulting heterozygous *MAD2* mutants were named ASR03M and ASR04M. The transformants were selected on CM-leu plates. The remaining alleles of *MAD2* in ASR03M and ASR04M were deleted by transformation with the plasmid pMad2-3 after digestion with BamHI-XhoI. The resulting *mad2* mutants from ASR03M and ASR04M were named ASR17 and ASR18, respectively. The transformants were selected on CM-leu-arg plates. The strains were confirmed by PCR using primer pairs Mad2-1/Mad2-2 and NV228/NV229.

### Construction of strains ASR19 through ASR21

To tag Ndc80 with mCherry, a previously reported plasmid pNdc80-mChARG (Varshney & Sanyal, 2019) was linearized with XhoI and used the transform strains ASR02 through ASR04 to obtain the strains ASR19 through ASR21. The transformants were selected on CM-arg plates. Expression of the fusion protein was confirmed by microscopy.

## Construction of *C. neoformans* strains

### Construction of strains CNSD169 through CNSD171

A 743-bp fragment that includes Dad2 promoter and the ORF was amplified by overlap PCR using the primer pairs SD118/SD119 and SD120/VYP152 to introduce the mutation R102A in CnDad2. By a similar overlap PCR-based strategy, The DSS-deleted version of CnDad2 was also amplified. A second fragment containing mCherry tag along with a neomycin marker was amplified from pLK25 using the primer pairs VYP153/VYP154. The homology region corresponding to the DAD2-3′UTR was amplified using VYP155/VYP156. These three fragments were subsequently fused by overlap PCR using the primer pairs SD118/VYP156 and used to transform the strain CNV108 by biolistic method. For the control strain, the same overlap PCR strategy was used with the first homology region amplified without any mutation using primer pairs SD118/VYP152. Transformants were selected by resistance to neomycin and were further confirmed by PCR and Sanger sequencing.

## Construction of *S. pombe* strains

### Construction of strains PSR01 and PSR02

The ORF of SpDad2 along with 534 bp of sequences upstream of Dad2 was amplified using the primer pairs SpF/Sp-FL-R. A DSS-deleted version of the same fragment was amplified using the primer pairs Sp-F/Sp-del-R, wherein the DSS deletion was introduced in the long primer. Both these primers were designed to have overhangs to facilitate their recombination into the plasmid pVB128 such that the ORF is in frame with the 3xFLAG tag from the plasmid. The resulting plasmids pSp-FL and pSp-del were then linearized with Nhe1 and used for transformation of the *S. pombe dad2* mutant strain SP1619 to obtain the strains PSR01 (expressing SpDad2-FL) and PSR02 (expressing SpDad2-ΔDSS). Standard LiOAc-PEG method was followed for the transformation and the transformants were selected by resistance to NAT. Targeted integration was verified by a PCR using the primers Sp-cFP/FLAG-RP.

## Identification of DSS

The HMM file for the Dad2 domain (PF08654) was downloaded from the Pfam database page for this domain. This model was used as the query to perform a HMMsearch (hmmsearch search | HMMER (ebi.ac.uk)) routinely used to search for a HMM profile against a protein sequence database. The search was targeted against fungal genomes using the option "Ensembl Genomes Fungi." The E-value cutoff for the search was 0.01 by default. The first round of HMM search resulted in 912 hits from all the major fungal subphyla (Table S5, Sheet- Hits_Ensembl Fungi). The hits were manually curated to remove duplicate entries of any given species for which multiple genome sequences were available. In addition to this, hits from nine species that were unusually long were also removed for subsequent analysis to avoid alignment artifacts (Table S5, sheet-long hits). This resulted in a final pool of 466 hits from which a multiple sequence alignment was generated using MAFFT. Apart from the known Dad2 domain at the N-terminus, the alignment revealed a second conserved domain Dad2 centered around a highly conserved arginine. This 10aa-long domain was named DSS. The amino acid logo indicating the consensus sequence of DSS (in Fig 1A) was generated with this multiple sequence alignment using Weblogo tool (WebLogo - Create Sequence Logos (berkeley.edu)).

## Flow cytometry

Overnight grown cultures of *S. cerevisiae* or *C. albicans* were subcultured to 0.2 $OD_{600}$ in desired media/growth condition as described for the experiment. Cells were harvested at various time intervals post-inoculation. Harvested samples were fixed by dropwise addition of ice-cold 70% ethanol, followed by RNAse treatment and propidium iodide (PI) staining as described before (Sanyal & Carbon, 2002; PNAS). Stained cells were diluted to the desired cell density in 1x PBS and analyzed (30,000 cells) by flow cytometry (FACSAria III; BD Biosciences) at a rate of 500–2,000 events/s. The output was analyzed using the FLOWJO software. The 561-nm laser was used to excite PI and 610/20 filter to detect its emission signals.

## Fluorescence microscopy and analysis

Cells were grown in media and growth conditions as described for each experiment. Before imaging, the cells were washed thoroughly with sterile water twice and resuspended in sterile water to achieve optimum cell density for imaging. The cells were imaged in Zeiss Axio observer equipped with Colibri 7 as the LED light source, 100x Plan Apochromat 1.4 NA objective, and PCO edge 4.2 sCMOS camera. Z sections were obtained at an interval of 300 nm. All the images were displayed after the maximum intensity projection using ImageJ. For visualizing nuclear mass, *S. cerevisiae* and *C. albicans* cells were stained with Hoechst (50 ng/ml) before imaging. GFP and mCherry fluorescence were acquired using the filter set 92 HE (excitation 455–483 nm and emission 501–547 nm for GFP, and excitation 583–600 nm, and emission 617–758 nm for mCherry).

## Western blotting

Protein lysates for Western blot were prepared by the TCA method. From overnight grown cultures, 3$OD_{600}$ equivalent cells were harvested, washed, and resuspended in 400 μl of 12.5% ice-cold TCA solution. The suspension was vortexed briefly and stored at −20°C for 12 h. The suspension was thawed on ice, pelleted at 14,000 rpm for 10 min, and washed twice with 350 μl of 80% acetone (ice cold). The washed pellets were air dried completely and resuspended in desired volume of lysis buffer (0.1N NaOH+1% SDS). Rabbit anti-Protein A antibody (P3775; Sigma-Aldrich) and the HRP-conjugated goat anti-rabbit secondary antibody, both were used at 1:5,000 dilution in 2.5% skim milk powder in 1XPBS. The blots were developed using chemiluminescent substrate (Bio-Rad) and imaged using Chemidoc system (Bio-Rad).

## Expression and purification Dam1 complex

### Plasmid construction

The plasmid co-expressing the 10 *S. cerevisiae* Dam1 complex subunits with a C-terminal His6-tagged DAD1 was used to generate the mutant versions DASH$^{R126A}$ and DASH$^{\Delta DSS}$ wherein the Dad2 ORF was mutated to encode Dad2-R126A and Dad2-ΔDSS. These changes were introduced as described below. We identified two unique sites NruI and SnaBI that contained the sequences encoding Dad1, the Dad2 and the N terminal part of Spc19. This fragment was amplified by overlap PCR using the primers mentioned in Table S4 such that the desired mutations were introduced in the Dad2 ORF. The resulting amplicons digested and cloned into the NruI/SnaBI sites of the plasmid PC4-Dad1H. The desired mutations were verified by sequencing.

### Expression and purification

The recombinant WT and the mutant *S. cerevisiae* Dam1 complexes were expressed and purified as described previously with some modifications (Miranda et al, 2005; Westermann et al, 2006). Briefly, the recombinant vector plasmids were transformed into *E. coli* BL21 Rosetta cells (Novagen) harboring the pRARE plasmid (Novagen). Super Broth media (3 liters) was inoculated with an overnight *E. coli* bacterial culture and grown at 37°C till $OD_{600}$ reached 0.4. The protein expression was induced by the addition of 0.3 mM IPTG. Bacterial cells were harvested after 4 h induction, washed with ice-cold PBS pH 7.5, and resuspended in lysis buffer (50 mM sodium phosphate [pH 7.5], 500 mM NaCl, 35 mM imidazole, and 5 mM β-mercaptoethanol). Cells were lysed by sonication (1 s ON, 1 s OFF, total 5 min, 40% amplitude), and the lysate was cleared by centrifugation at 22,000 rpm (JA 25.50; Beckman Coulter) at 6°C for 50 min. The supernatant was incubated with preequilibrated Ni-NTA Agarose (QIAGEN) beads under constant rotation at 4°C for 2 h. After incubation, beads were recovered by low-speed centrifugation at 500*g* for 5 min, washed with 10-column volumes of lysis buffer (once), high-salt buffer (twice, 50 mM sodium phosphate [pH 7.5], 500 mM NaCl, 35 mM imidazole, 50 mM KCl, 10 mM MgCl 2, 2 mM ATP, and 5 mM β-mercaptoethanol). The proteins were eluted by incubating with 40 ml of elution buffer (50 mM sodium phosphate [pH 7.5], 500 mM NaCl, 400 mM imidazole, and 5 mM β-mercaptoethanol) and dialyzed overnight against 2 liters of dialysis buffer (50 mM sodium phosphate [pH 7.5], 25 mM NaCl, 1 mM EDTA, and 1 mM β-mercaptoethanol). The dialyzate was subsequently loaded onto

a preequilibrated 5 ml HiTRAP Q HP(Cytiva) ion exchange column, and the bound proteins were eluted in 3-ml fractions by a linear gradient of 15–1,000 mM NaCl. The fractions containing all the 10 Dam1 complex subunits were pooled, concentrated using a VIVASPIN 15R 10,000 MWCO (Sartorius) centricon, and loaded to a Superose 6 10/300 GL column (Cytiva). The bound proteins were eluted with 25 mM HEPES pH 7.4 500 mM NaCl 1 mM EDTA. The purified protein samples were concentrated to 1 mg/ml, resuspended to a 10% glycerol, and stored in –80°C as 100 $\mu$l aliquots until further use.

### MT polymerization, co-sedimentation, and electron microscopy

MT polymerization and co-sedimentation assays were performed according to a previously reported procedure (Miranda et al, 2005) with minor modifications. Briefly, bovine tubulin (Cytoskeletons Inc.) was polymerized at 5 mg/ml in 80 mM PIPES pH 6.8, 1 mM GTP, 1 mM MgCl 2, 1 mM EGTA, 100 $\mu$M paclitaxel at 35°C for 20 min. The paclitaxel-stabilized MTs were diluted 100-fold in 25 mM HEPES, 100 mM NaCl, 1 mM GTP, and 10 $\mu$M paclitaxel.

### MT-decoration with Dam1 complex

The purified Dam1 complex aliquots were thawed on ice and precleared by centrifugation at 20,000$g$ for 10 min at 4°C. About 20 $\mu$l of 1 mg/ml Dam1 complex was mixed with 180 $\mu$l of MT solution and incubated for 20 min at room temperature.

For MT co-sedimentation assay, the Dam1 complex–MT reaction mix was gently layered onto 250 $\mu$l cushion buffer (25 mM HEPES, 100 mM NaCl, 50% glycerol, 1 mM EGTA, 1 mM MgCl$_2$, 10 $\mu$M paclitaxel, 4 mM DTT) and centrifuged (TLA 100.3 rotor) at 80,000 rpm for 10 min at 25°C. The supernatant (top 150 $\mu$l) and pellets (resuspended in 220 $\mu$l 1x loading buffer) were recovered and analyzed on 15% SDS–PAGE. Gels were stained with Coomassie solution, destained, and photographed.

For visualization of Dam1-MT decoration, 4 $\mu$l of the Dam1 complex–MT reaction mix was adsorbed onto a freshly glow-discharged (25 mA, 0.38 mbar, 60 s) grid for 2 min, washed with 25 mM HEPES pH 7.4 500 mM NaCl 1 mM EDTA buffer, and stained with 0.7% (wt/vol) uranyl formate. Micrographs of negatively stained samples were captured using the Joel JEM1400 Plus TEM instrument (University of Edinburgh).

### Statistical analysis

All graphs were generated and analyzed using GraphPad Prism (v8). One-way ANOVA or paired $t$ test with Welsch's correction was used based on the sample size to test for the statistical significance of acquired results using a $P$-value cut off of 0.05.

## Supplementary Information

## Acknowledgements

We thank S Biggins (Fred Hutchinson Cancer Research Center, USA) for sharing the *S. cerevisiae* strain SBY12503, A Marston (University of Edinburgh, UK) for sharing *S. pombe* strains 22 and 1619, and V Borde (Institut Curie, France) for sharing the plasmid pVB128. The details are included in the list of strains and plasmids. We thank the in-house flow cytometry facility at JNCASR, and Dr. N Nala for assistance with flow cytometric analysis. We thank M Alba Abad for helping with the purification of Dam1 complexes and MT binding assays, and A Gireesh for assistance with negative staining for EM. We also thank the CTCB facility manager Martin A Wear (University of Edinburgh) for assistance with SEC-MALS analysis of Dam1 complex. We also thank TK Manna and Renjith MR (IISER, Thiruvananthapuram), H Balaram and A Bellur (JNCASR, Bangalore), and KVR Chary (IISER, Berhampur) for useful discussions and feedback. We are grateful for the support from Ines Drinnenberg (Institut Curie, France) during the revision of this article. SR Sankaranarayanan was a research associate supported by intramural funding by JNCASR. SD Polisetty and K Das are senior research fellows supported by the Graduate Research Fellowship from JNCASR. K Das acknowledges the EMBO Scientific Exchange Grant (9809) for support received to visit AA Jeyaprakash's laboratory for biochemical characterization of Dam1 complex. AA Jeyaprakash is supported by the Wellcome Trust through a Senior Research Fellowship (202811). AA Jeyaprakash and his team are co-funded by the European Union (ERC, CHROMSEG, 101054950). Views and opinions expressed are, however, those of the author(s) only and do not necessarily reflect those of the European Union or the European Research Council. Neither the European Union nor the granting authority can be held responsible for them. K Sanyal is a JC Bose National Fellow (JCB/2020/000021) of Science and Engineering Research Board (SERB), Department of Science and Technology, Govt. of India. A Dumbrepatil was a research associate supported by JNCASR. K Sanyal acknowledges financial support from the Department of Biotechnology, Govt. of India and SERB, and intramural funding from JNCASR.

### Author Contributions

SR Sankaranarayanan: conceptualization, data curation, formal analysis, investigation, visualization, methodology, and writing—original draft, review, and editing.
SD Polisetty: formal analysis, investigation, visualization, and methodology.
K Das: formal analysis, investigation, visualization, methodology, and writing—review and editing.
A Dumbrepatil: formal analysis, investigation, and methodology.
B Medina-Pritchard: investigation, visualization, and methodology.
M Singleton: investigation, visualization, and methodology.
AA Jeyaprakash: supervision, funding acquisition, validation, visualization, methodology, project administration, and writing—review and editing.
K Sanyal: conceptualization, resources, supervision, funding acquisition, methodology, project administration, and writing—original draft, review, and editing.

### Conflict of Interest Statement

The authors declare that they have no conflict of interest.

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
