## [Reviewer comments · Life Science Alliance]

Life Science Alliance

Functional Plasticity in Chromosome-Microtubule Coupling on the Evolutionary Time Scale

Sundar Sankaranarayanan, Satya Dev Polisetty, Kuladeep Das, Arti Dumbrepatil, Bethan Medina-Pritchard, Martin Singleton, A. Jeyaprakash, and Kaustuv Sanyal

DOI: <https://doi.org/10.26508/lsa.202201720>

Corresponding author(s): Kaustuv Sanyal, Jawaharlal Nehru Centre for Advanced Scientific Research

Review Timeline:	Submission Date:	2022-09-12
	Editorial Decision:	2022-10-19
	Revision Received:	2023-07-31
	Editorial Decision:	2023-09-08
	Revision Received:	2023-09-15
	Accepted:	2023-09-19

Transaction Report:

October 19, 2022

Re: Life Science Alliance manuscript #LSA-2022-01720-T

Dr. Kaustuv Sanyal
Jawaharlal Nehru Centre for Advanced Scientific Research
Molecular Biology and Genetics Unit
Molecular Mycology Laboratory
Bangalore, KA 560064
India

Dear Dr. Sanyal,

Thank you for submitting your manuscript entitled "Acquired functional redundancy of an invariant amino acid residue in an evolutionarily conserved protein family" to Life Science Alliance. The manuscript was assessed by expert reviewers, whose comments are appended to this letter. We invite you to submit a revised manuscript addressing the Reviewer comments.

Thank you for this interesting contribution to Life Science Alliance. We are looking forward to receiving your revised manuscript.

Sincerely,

B. MANUSCRIPT ORGANIZATION AND FORMATTING:

Reviewer #1 (Comments to the Authors (Required)):

The manuscript "Acquired functional redundancy of an invariant amino acid residue in an evolutionary conserved protein family" by Sankaranarayanan and colleagues describes a genetics study that explores the role of a conserved arginine residue in the Dam1 subunit Dad2 in kinetochore plasticity. Analyzing the fungi-specific outer kinetochore protein Dad2 from over 500 fungi species, a new conserved domain was found, which the authors named DDS (Dad2 signature sequence), which includes a very well conserved arginine residue. From the budding yeast crystal structure of the DAM1 complex, a structural role for Dad2 DDS domain was hypothesized. This was subsequently tested. Several mutants were made in three yeast species: *S. cerevisiae* (Sce), *C. albicans* (Cal), and *C. neoformans* (Cne) where either the conserved or less conserved arginine was mutated to an alanine, or the DDS motif was deleted altogether. Subsequently, the spindle pole, spindle length, chromosome segregation errors, and cell cycle progression were assessed. Where in both Sce and Cal, DDS and R116/92 mutants showed defects in all tested parameters, Cne DDS and R102 mutant did not show growth defects, but did show a small increase in large bud arrested cells when DDS was deleted. The authors argue that the size difference of the functional CENP-A domain is at the core of this differential observation. In Sce and Cal, only one CENP-A/Cse4 is bound by a single microtubule (MT), where the much larger regional centromere of Cne binds multiple MTs per centromere. This is an enticing model that can be further supported by some of the concerns mentioned below. Overall, this is a very nice and well written evolutionary study on the importance of a single conserved residue in cell division.

Major concerns:

- + The authors mention in the introduction that overexpression of CENP-A in Cal results in increased CENP-A nucleosomes at the centromere, subsequently recruiting more outer kinetochore components (Burrack et al 2011 Curr Biol). Alternatively, the engineered Sce that can accommodate additional MTs (Nannas et al 2014 MBoC) is another option. Either of these two models would be the ideal direct test to transform a DDS-sensitive species into a DDS-less sensitive or even DDS-insensitive species.
- + Figure 1a - the DDS motif shown underneath the sequence alignment appears to have an additional residue between the first P114 and the third Q/5114, as the sequence alignment does not have any residue positioned between P114 and Q/E115. This makes the conservation logo hard to interpret based on the shown sequence alignment.
- + The conservation of DDS is nicely shown and its potential role in Dam1 structure is highlighted. One does wonder how well the interacting residues of Spc19, Spc34, and Dam1 are conserved. If this were an important 'central domain' for the structure and function of Dam1, one would expect the DDS-interacting residues to be also highly conserved. In addition, by co-IP it could be shown if the mutant Dad2 indeed has impaired interactions with the other three Mad1 complex components. These data would add informative evolutionary and biochemical confirmation of the hypothesized structural implications of the DDS motif.
- + It would be of added values if the flow cytometry data was quantified, in addition to the qualitative graphs of Figures 2B and 3B.
- + The statement that "our study highlights the progressive loss of essentiality of a conserved arginine residue in Dad2" is misleading. This would imply that the point centromere is the default type of centromere, and not regional centromeres. This is not supported by the phylogenetic distribution of point centromeres. Point centromeres appear to be a highly specialized form of centromere, derived from regional centromeres. If anything, the opposite could be argued. That there is progressive gain of essentiality of a conserved arginine residue in Dad2 when a species evolves towards a single MT binding per centromere.

Minor concerns:

- + In figures 2 and 3, flow cytometry plots are shown to represent the cell cycle progression. For people who are not familiar with the yeast cell cycle and ploidy, it might be confusing to only 1N and 2N. Maybe it would be more informative for the non-yeast reader if G1 and G2/M were used to label the two peaks.
- + In figure 2b, the top row of histograms represents the 8hour mark, whereas the bottom row represents the beginning mark (0hour). It might be more intuitive for the reader if the top row was the beginning time point and the 8hr row is found below that.
- + Figure 3 - on the right-hand side are the labels for the two *Candida* species. It might be easier for the reader to grasp if these labels were placed above the section, instead of on the side.
- + Figure 3H/I - it would be informative for the reader if example images were shown to better understand what the bar-graphs represent.
- + In Figure 1C/F, the number of the residues are noted, but not in Figure 3F. For continuity it would be nice to see also residue

numbers in Figure 3F.

+ In Figure 4C, it would be helpful if the median with 25 and 75 percentiles were shown as well as the mean to allow the reader to more easily see the trend implied by the large number of data points. In addition, the positioning of this figure is confusing. The rest of the figures compared the three species used in this study, whereas 4C only looks at *Cal*. It would be a much stronger figure if the Ndc80-mCherry was used to compare the three species. The interpretation of the data as described in the discussion at page 20 (bottom first paragraph) is not accurate. Increased Ndc80 intensity does not per se mean recruitment of multiple MTs. It could also mean that a larger outer kinetochore is built to more effectively capture a single MT.

Reviewer #2 (Comments to the Authors (Required)):

The authors identify a highly conserved domain in the Dad2 subunit of the fungal-specific outer kinetochore complex DASH/Dam1 using multiple sequence alignment analysis. They use simple but effective genetics approaches and fluorescence microscopy to show that this domain (DSS), and in particular a conserved arginine residue, R126, is important for proper function of Dad2 in two ascomycota species, *S. cerevisiae* and *C. albicans*, organisms with point and regional centromeres, respectively. They find that DSS is essential for growth in *S. cerevisiae* but not in *C. albicans*. They then choose to study a more distantly related basidiomycota species, *C. neoformans*, which has a larger regional centromere, and find that DSS is not essential for growth in this species and has a "milder" phenotype associated with DSS mutants. The authors present a well-thought-out model concluding that the tolerance towards the mutation of the conserved arginine in DSS depends on the centromere length in these species. However, these conclusions are not sufficiently supported by the data presented and require substantial additional work to be convincingly justified.

Major points

1) The title needs to be more descriptive of the work and should include the DASH/Dam1 complex, which is the main subject of the research.

Although the authors present data showing an important conserved protein domain of Dad2 that is essential for accurate chromosome segregation in *S. cerevisiae* and *C. albicans*, the interest is limited, and it is known in the field that fungal species with regional centromeres are less reliant on the DASH/Dam1 complex compared to *S. cerevisiae*. Their experimental approaches are suitable for the work and the methodology is sound. However, they rely heavily on indirect evidence and their main argument that the functional significance of DSS is dependent on centromere length is highly speculative and not convincing for the following reasons:

2) The direct experimental evidence supporting the conclusions is limited and the main argument relies heavily on correlational observations. The authors' conclusion would be more convincing if they attempted to reduce the effects of confounding factors from evolutionary distance by studying additional species both with similar and different centromere lengths. For instance, another distantly related species with a point centromere, ascomycota species with a large regional centromere, and basidiomycota with a small regional centromere. In other words, if a more closely related species with a large regional centromere would be less sensitive to the DSS mutations than a more distantly related species with a small centromere, the authors' argument would be more persuasive.

3) Sequence alignment analysis of other DSS-interacting protein domains of neighbouring proteins is lacking. It is important for the reader to know whether or not the residues that the authors point out in Fig. 1B and Fig. S3 (for example residues E50 and I52 of Spc19), that interact with DSS or specifically the arginine R126 of Dad2 are also highly conserved.

4) Related to the point above, a simple structural investigation (similar to Fig. 1B and Fig. S3) of the species-specific DASH/Dam1 subunits neighbouring DSS in each of the three species studied, for example by using tools such as AlphaFold (<https://alphafold.ebi.ac.uk/>), may highlight any potential differences of the DSS interface in the different species.

5) The basis of the authors' claims regarding centromeric chromatin length, ability to bind multiple microtubules and the progressive loss of essentiality of the conserved arginine of Dad2 is not convincing. For example, the authors point out (Fig. 4B legend and Fig. S4A) that *C. neoformans* centromere's ability to bind multiple microtubules is predicted from the size of the centromere, but this is not verified. Furthermore, the authors also point out (page 20 bottom) that they expect at least 10 CENPA nucleosomes to be present on each *C. neoformans* centromere based on the regional centromere length of other species, which may be true, however as mentioned by the authors some large regional centromeres only bind about 2-3 microtubules. Based on the importance of these details for the authors' main argument, these points need to be addressed further and verified experimentally, and not simply "predicted and expected".

6) The experiments performed to investigate the effects of DSS mutants in *C. neoformans* are inadequate for comparing results across the three species studied. For *S. cerevisiae* and *C. albicans*, the authors adequately show that DSS mutants affect cell-cycle progression, spindle dynamics, and timely nuclear segregation. However, they fail to address this for the species they choose to compare these phenotypes with. At the very least, they could have analysed large-budded cells at different temperatures, since this was done for *C. albicans* and showed an increased phenotype at a lower temperature. Further

characterization of the mutants in *C. neoformans* in line with what was done for the other species is crucial for a satisfactory comparison.

7) The authors claim that kinetochore assembly is not affected by the DSS mutations in *C. albicans* and *C. neoformans*, however this is not directly addressed. The authors speculate that this domain may contribute to the overall structure of the DASH/Dam1 complex in *S. cerevisiae*, and since subunits of the DASH/Dam1 complex are not essential in some yeasts, it would be important to know whether these mutants affect the complex structurally. The authors show that the fluorescently labelled dad2-R126A and dad2-R92A mutants in *S. cerevisiae* and *C. albicans* can localise correctly and form foci, but they fail to quantify the fluorescence intensity of the foci, which should be an easy additional analysis for the authors, and especially important since they do show that ScDad2- Δ DSS mutant fluorescence signal is diffused and thus it is possible that the arginine mutants have a lower intensity, which would suggest a compromised kinetochore assembly.

Minor points

Representative micrographs of the fluorescence microscopy analyses in Figs. 2C, 3D, 3E, 3H, 3I and 4C, either in the main or supplementary figures would be good.

Page 5, line 8 from bottom: It would help to see the amino acid conservation of the interacting residues from the other DASH/Dam1c proteins within the 'central domain' (see major point 3).

Page 6, Fig. 1A: The residue numbers above sequence alignment belong to which species?

Page 7, Fig. 1B legend: the authors should point out here that the structure is based on DASH/Dam1 from another species *Chaetomium thermophilum* as they do in Fig. S3.

Page 8, chapter title and Fig.1 caption: Even though the authors have shown that DSS is important for *S. cerevisiae* Dad2 and less so for *C. albicans*, they have not provided sufficient data in this chapter to conclude that DSS function is "inversely correlated to the centromere length", thus the chapter heading should be adjusted. Similarly, the Fig.1 caption stating that DSS function is "dependent on the length of centromeric chromatin" should be changed (also see major points 2-6).

Page 9, line 16 from top: The authors state that they "did not observe any significant growth retardation" of ASR03 and ASR06 at 30°C, however it is clear in Fig. 1G that growth is inhibited at 30°C although not to the same extent as 18°C.

Page 9, line 11 from bottom (related to point above): Based on the growth inhibition at 30°C, the authors' conclusions should be adjusted.

Page 10, line 16 from top: Citation missing after "...checkpoint (SAC)"

Page 10, line 4 from bottom: The word "colocalized" is not correctly used here. The authors analyse GFP foci between two Spc110-mCherry foci and measure the distance between SPBs and quantify monooriented spindle phenotype. The word is also incorrectly used in figure legends Fig. 2D.

Page 11, 1st sentence: Reference to Fig. 2D and 2E is missing

Page 11, lines 3-5 from top: This whole section is unclear and difficult to read

Page 11, Figure 2 legend: The reader does not know what "SPB" is at this stage in the manuscript and should be explained.

Page 12, line 17 from bottom: The authors sometime use "CENPA" and sometimes "Cse4" for *C. albicans*, more consistency throughout the manuscript would help readers that are not familiar with the field.

Page 12, line 15 from bottom: The authors claim the "kinetochore integrity is not compromised", however this possibility has not been ruled out. For example, quantification of fluorescence intensity of the Dad2-GFP WT and mutants would be beneficial (see major point 7).

Page 12, line 13 from bottom: Should be Fig. 1H not 1G. Similar to point above, the authors have not directly investigated kinetochore assembly and further analysis of the kinetochore signal is needed to rule out that deletion of DSS does not affect kinetochore assembly, especially since both the dad2-R92A and dad2- Δ DSS cause growth defects at 30°C and 18°C (Fig. 1G).

Page 12, the bottom 3 lines: Again, related to the points above, the effects of DSS mutants on kinetochore assembly cannot be ruled out based on the data presented. Furthermore, effects on kinetochore assembly and kinetochore-microtubule interactions are two possibilities that are not mutually exclusive.

Page 12, line 3 from bottom: Should be Fig. 1H not 1G

Page 13, Figure 3 I: The Y axis should be corrected to "unsegregated nucleus" as in Fig. 3C not "stuck nucleus".

Page 14, line 6 from bottom: The authors state that they observe "increase in the frequency of defective nuclear segregation", however they do not directly show this experimentally, and therefore should use the phrase "unsegregated nuclei" as they do in the figure and figure legends (Fig. 3C).

Page 14, line 2 from bottom: Same point as above.

Page 14, line 3 from bottom: "SPB" has not been explained in the main text at this stage in the manuscript.

Page 15, line 14 from bottom: "mother bud" should be corrected to "mother cell"

Page 15, line 4 from bottom: Citation missing after "... *C. albicans*"

Page 16, line 13 from bottom: Reference to Fig. 3H is not needed here.

Page 16, line 3 from bottom: What do the authors mean by "wild-type-like behaviour"? This sentence needs more clarification.

Page 17, line 2 from top: The authors' conclusion that the "dependence on the conserved arginine residue for mitotic progression across the evolutionary diverged species can be attributed to the differences in the centromeric length" is speculative and the data presented are not strong enough to support this (see major points 2-6).

Page 17, line 17 from bottom: The authors do not show any analysis of "rate of chromosome missegregation" in this manuscript, thus this sentence should be removed or modified.

Page 18, Figure 4C: I'm not sure about LSA policy, but typically new experiments should not be presented for the first time in the discussion, therefore Fig. 4C and the analysis should be relocated to the results section.

Page 20, line 5-6 from bottom: This sentence is unclear and I'm not sure what the authors mean by "...even in wild-type cells?"

Page 21, line 11-12: The sentence is unclear

Page 26, line 3 from bottom: Space missing between "to" and "0.2"

Page 33, 2nd reference from bottom: Author names are missing.

Reviewer #3 (Comments to the Authors (Required)):

The study focuses on Dad2 a protein of the ring complex that is essential to tether kinetochores to microtubules. The authors show that a residue of Dad2 R126 is essential in *S.cerevisiae* but not in *Candida albicans* or *Cyptococcus neoformans*. They propose that the tolerance is because of regional and not point kinetochores in *c. albicans* and *c. neoformans*. This is an exciting hypothesis positioning regional centromeres as a fail-safe mechanism which allows tolerance to lethal mutations. The experiments are well done. I would recommend the article for publication with very minor changes.

1. The experiments are well done and the authors present clear evidence for the lack of essentiality of the R128 amino acid in CnDad2. But this is only a correlation with differences in the length of centromeres. So to indicate this clearly, I would recommend the authors clearly double-check their interpretation as a correlation and not as a causal link.

The last sentence of the result (p17) suggests it as a causal link and this should be reworded: "dependence on this conserved arginine residue for mitotic progression across the evolutionarily diverged species can be attributed to the differences in the centromere length."

2. The discussion and introduction section clearly document that the manuscript is about a single nucleotide in Dad2. However, the abstract opens with a very wide question which could seem to distract the reader from the original findings.

3. Fig 3C requires scale bars and mentioning the inversion of intensities

Reviewer #1 (Comments to the Authors (Required)):

The manuscript "Acquired functional redundancy of an invariant amino acid residue in an evolutionary conserved protein family" by Sankaranarayanan and colleagues describes a genetics study that explores the role of a conserved arginine residue in the Dam1 subunit Dad2 in kinetochore plasticity. Analyzing the fungi-specific outer kinetochore protein Dad2 from over 500 fungi species, a new conserved domain was found, which the authors named DDS (Dad2 signature sequence), which includes a very well conserved arginine residue. From the budding yeast crystal structure of the DAM1 complex, a structural role for Dad2 DDS domain was hypothesized. This was subsequently tested. Several mutants were made in three yeast species: *S. cerevisiae* (Sc), *C. albicans* (Ca), and *C. neoformans* (Cn) where either the conserved or less conserved arginine was mutated to an alanine, or the DDS motif was deleted altogether. Subsequently, the spindle pole, spindle length, chromosome segregation errors, and cell cycle progression were assessed. Where in both Sc and Ca, DDS and R116/92 mutants showed defects in all tested parameters, Cn DDS and R102 mutant did not show growth defects, but did show a small increase in large bud arrested cells when DDS was deleted. The authors argue that the size difference of the functional CENP-A domain is at the core of this differential observation. In Sc and Ca, only one CENP-A/Cse4 is bound by a single microtubule (MT), where the much larger regional centromere of Cn binds multiple MTs per centromere. This is an enticing model that can be further supported by some of the concerns mentioned below. Overall, this is a very nice and well written evolutionary study on the importance of a single conserved residue in cell division.

Major concerns:

The authors mention in the introduction that overexpression of CENP-A in Ca results in increased CENP-A nucleosomes at the centromere, subsequently recruiting more outer kinetochore components (Burrack et al 2011 Curr Biol). Alternatively, the engineered Sc that can accommodate additional MTs (Nannas et al 2014 MBoC) is another option. Either of these two models would be the ideal direct test to transform a DDS-sensitive species into a DDS-less sensitive or even DDS-insensitive species.

- As pointed out by the reviewer, Burrack et al.,(2011) indeed showed that overexpression of CENPA resulted in increase in the number of CENPA nucleosomes, and other outer kinetochore proteins at the centromeres. In such a scenario, they find the Dam1 complex subunits to be less essential as compared to a scenario with wildtype CENPA levels. The fact that the depletion of Dam1 complex subunits itself could be tolerated in CENPA overexpression conditions suggested that the effect of deleting/mutating DSS could be insignificant in this background.
- Regarding the results from Nannas et al.: We would like to point out that in the engineered strains developed in their study, they were able to maintain additional copies of centromeric plasmids due to the ability of engineered strain to form more kMTs in the cell. However, the number of kMTs/ chromosome in this system remains 1 due to the point centromere configuration. We raised the same points in our discussion to explain why *S. cerevisiae* cells are unable to tolerate mutation/deletion of DSS despite being able to form around 50 additional kMTs in a cell (not per chromosome). This has been described in the discussion (P17 onwards)- "The lack of such a mechanism in *S. cerevisiae* despite these abilities can be attributed to their point centromere structure. The size restriction and dependence on DNA sequence prevent the enrichment or spreading of kinetochore proteins at the native centromere, thereby impeding the formation of additional MT attachments."

Figure 1a - the DDS motif shown underneath the sequence alignment appears to have an additional residue between the first P114 and the third Q/5114, as the sequence alignment does not have any residue positioned between P114 and Q/E115. This makes the conservation logo hard to interpret based on the shown sequence alignment.

The additional residue pointed out by the reviewer is due to the presence of an additional amino acid in this position in few of the 466 Dad2 sequences analyzed in this study. The amino acid logo was generated from this alignment. However, we included a representative alignment in figure 1A to emphasize the conservation of DSS and the arginine residue in it (red colored, position 119 in the alignment) across different centromere structures. This has been mentioned in the legend for figure 1.

The conservation of DDS is nicely shown and its potential role in Dam1 structure is highlighted. One does wonder how well the interacting residues of Spc19, Spc34, and Dam1 are conserved. If this were an important 'central domain' for the structure and function of Dam1, one would expect the DDS-interacting residues to be also highly conserved. In addition, by co-IP it could be shown if the mutant Dad2 indeed has impaired interactions with the other three Mad1 complex components. These data would add informative evolutionary and biochemical confirmation of the hypothesized structural implications of the DDS motif.

As suggested by the reviewer, we have included a sequence alignment highlighting the conservation of the Dad2-interacting regions in Spc19.

We further validated the significance of these contacts by purifying the Dam1 complex that contains Dad2-R126A or Dad2-ΔDSS instead of wild-type Dad2. We find that Spc19 fails to copurify with the complex in the presence of Dad2-R126A. These findings have been described in Figure 3 and the corresponding results section.

It would be of added values if the flow cytometry data was quantified, in addition to the qualitative graphs of Figures 2B and 3B.

The indicated flow cytometry data has been quantified and presented in figure S2F.

The statement that "our study highlights the progressive loss of essentiality of a conserved arginine residue in Dad2" is misleading. This would imply that the point centromere is the default type of centromere, and not regional centromeres. This is not supported by the phylogenetic distribution of point centromeres. Point centromeres appear to be a highly specialized form of centromere, derived from regional centromeres. If anything, the opposite could be argued. That there is progressive gain of essentiality of a conserved arginine residue in Dad2 when a species evolves towards a single MT binding per centromere.

We concur with the reviewer that the point centromeres are an evolved state and thank for pointing out the ambiguity. We merely draw an inverse correlation between centromere length and the significance of this motif. We make no inferences on the ancestral centromere state. We have clarified this in the

revised manuscript by rephrasing this sentence (P18)-“ In this study, we uncover another potential advantage associated with regional centromeres in tolerating conditions sub-optimal for chromosome segregation. This adaptive edge provided by regional centromeres in enabling compensatory mechanisms may explain their recurring presence across Eukaryota (Figure S7B, Table S1). Our results also highlight a significant role of centromere size/length in shaping the evolution of Dam1 complex function and its physiological structure. This can be attributed to several features such as the presence of excess CENPA nucleosomes, tethering multiple kMTs per chromosome, and/or enabling multivalency of Ndc80. The ring structure could be an evolutionarily selected conformation that is indispensable for the attachment of chromosomes with point centromeres, a derived state that provides a single attachment point to the MTs.”.

Minor concerns:

+ In figures 2 and 3, flow cytometry plots are shown to represent the cell cycle progression. For people who are not familiar with the yeast cell cycle and ploidy, it might be confusing to only 1N and 2N. Maybe it would be more informative for the non-yeast reader if G1 and G2/M were used to label the two peaks.

We have modified the labels as suggested by the reviewer.

+ In figure 2b, the top row of histograms represents the 8hour mark, whereas the bottom row represents the beginning mark (0hour). It might be more intuitive for the reader if the top row was the beginning time point and the 8hr row is found below that.

We have rearranged the FACS histograms as suggested by the reviewer.

+ Figure 3 - on the right-hand side are the labels for the two *Candida* species. It might be easier for the reader to grasp if these labels were placed above the section, instead of on the side.

We have modified the label as suggested.

+ Figure 3H/I - it would be informative for the reader if example images were shown to better understand what the bar-graphs represent.

Representative images have been included adjacent to the graphs in the revised manuscript.

+ In Figure 1C/F, the number of the residues are noted, but not in Figure 3F. For continuity it would be nice to see also residue numbers in Figure 3F.

We have included the residue numbers in the revised manuscript (Figure 4 in the revised version).

+ In Figure 4C, it would be helpful if the median with 25 and 75 percentiles were shown as well as the mean to allow the reader to more easily see the trend implied by the large number of data points. In addition, the positioning of this figure is confusing. The rest of the figures compared the three species used in this study, whereas 4C only looks at *Cal*. It would be a much stronger figure if the Ndc80-mCherry was used to compare the three species. The interpretation of the data as described in the discussion at page 20 (bottom first paragraph) is not accurate. Increased Ndc80 intensity does not per se mean recruitment of multiple MTs. It could also mean that a larger outer kinetochore is built to more effectively capture a single MT.

- Figure 4C has been modified as suggested by the reviewer (Figure 6C in revised manuscript).
- Regarding the suggestion to use Ndc80-mCherry for all the three species: The measurements made for Figure 6C were from a population of mutants (either R92A, or DSS deletion) that were still able to segregate their chromosomes despite having the mutation. The lethality of these mutations in *S. cerevisiae* obscured the use of this assay. Unlike *C. albicans* which requires additional outer

kinetochore proteins to tether additional kMTs per chromosome, the large regional centromeres of *C. neoformans* and *S. pombe* can bind multiple kMTs per chromosome even in a wild-type scenario. Further, DSS mutants in both these species do not show any defects in physiological or stress conditions (ex: low temperature growth or presence of thiabendazole). Therefore, we do not suspect elevated levels of Ndc80 upon mutations in the DSS in this species. We therefore performed this assay only in *C. albicans*.

- We concur with the reviewer's comment that an increase in Ndc80 levels does not directly represent recruitment of multiple kMTs. In the revised manuscript, we include and discuss the possibility of the requiring more Ndc80 to efficiently capture a single kMT upon mutating Dad2 in *C. albicans* (P17)- "We also propose an alternate, non-mutually exclusive possibility wherein the enrichment of Ndc80 by itself can facilitate efficient MT capture by a weakened kinetochore. Several studies have shown that Ndc80 can form crosslinks with adjacent Ndc80 molecules through the unstructured loop region (Maure et al. 2011; Polley et al. 2023). The consequence of such homotypic crosslinking is a cooperative increase in the affinity to MTs binding and the ability to stay associated with MTs under tension. This translates to the increased stability of end-on attachments in vivo (Hsu and Toda 2011; Tang et al. 2013; Helgeson et al. 2018; Polley et al. 2023). Despite the unstructured loop being conserved across eukaryotes, the loop-mediated multivalency of Ndc80 molecules can have a stronger impact on MT binding in large regional centromeres as compared to point centromeres due to the possible limits of Ndc80 enrichment that can be accommodated."

Reviewer #2 (Comments to the Authors (Required)):

The authors identify a highly conserved domain in the Dad2 subunit of the fungal-specific outer kinetochore complex DASH/Dam1 using multiple sequence alignment analysis. They use simple but effective genetics approaches and fluorescence microscopy to show that this domain (DSS), and in particular a conserved arginine residue, R126, is important for proper function of Dad2 in two ascomycota species, *S. cerevisiae* and *C. albicans*, organisms with point and regional centromeres, respectively. They find that DSS is essential for growth in *S. cerevisiae* but not in *C. albicans*. They then choose to study a more distantly related basidiomycota species, *C. neoformans*, which has a larger regional centromere, and find that DSS is not essential for growth in this species and has a "milder" phenotype associated with DSS mutants. The authors present a well-thought-out model concluding that the tolerance towards the mutation of the conserved arginine in DSS depends on the centromere length in these species. However, these conclusions are not sufficiently supported by the data presented and require substantial additional work to be convincingly justified.

Major points

1) The title needs to be more descriptive of the work and should include the DASH/Dam1 complex, which is the main subject of the research.

Although the authors present data showing an important conserved protein domain of Dad2 that is essential for accurate chromosome segregation in *S. cerevisiae* and *C. albicans*, the interest is limited, and it is known in the field that fungal species with regional centromeres are less reliant on the DASH/Dam1 complex compared to *S. cerevisiae*. Their experimental approaches are suitable for the work and the methodology is sound. However, they rely heavily on indirect evidence and their main argument that the functional significance of DSS is dependent on centromere length is highly speculative and not convincing for the following reasons:

2) The direct experimental evidence supporting the conclusions is limited and the main argument relies heavily on correlational observations. The authors' conclusion would be more convincing if they attempted to reduce the effects of confounding factors from evolutionary distance by studying additional species both with similar and different centromere lengths. For instance, another distantly related species with a point centromere, ascomycota species with a large regional centromere, and basidiomycota with a small regional centromere. In other words, if a more closely related species with a large regional centromere would be less sensitive to the DSS mutations than a more distantly related species with a small centromere, the authors' argument would be more persuasive.

We restricted to the three species used in this study for the following reasons.

The Saccharomycotina is the only clade with species that possess point centromeres. There is no second origin of point centromeres besides this clade in fungi. Thus, we used *S. cerevisiae* as a representative of point centromeres. The shortest centromeres known in the Basidiomycota are from the *Malassezia* species (small regional centromere type). Besides the lack of genetic tools, no other mitotic features have been experimentally studied in this species. Thus, we were unable to have a representative point/small regional centromere from Basidiomycota. Of the species with large regional centromeres, we chose *C. neoformans* because the Dam1 complex subunits are essential for viability, like the other yeasts analyzed in this study. This can be an appropriate background over which we can compare the role of DSS/ conserved arginine and make inferences therein.

We acknowledge the evolutionary distance between the species representative of the three centromere types and have included results from *S. pombe*, an ascomycete with large regional centromere. We find DSS to be dispensable in *S. pombe* as well. The relevant data has been included in Figure 5 in the revised manuscript.

SankaranarayananSR_ Fig5

3) Sequence alignment analysis of other DSS-interacting protein domains of neighboring proteins is lacking. It is important for the reader to know whether or not the residues that the authors point out in Fig. 1B and Fig. S3 (for example residues E50 and I52 of Spc19), that interact with DSS or specifically the arginine R126 of Dad2 are also highly conserved.

In the revised manuscript, we add supporting evidence to show the degree of conservation of the residues in Spc19 that can interact with the conserved arginine (Figure S1, added here in response to R1-major comment 3). While we do see substitutions, the charge is conserved in the position.

4) Related to the point above, a simple structural investigation (similar to Fig. 1B and Fig. S3) of the species-specific DASH/Dam1 subunits neighbouring DSS in each of the three species studied, for example by using

tools such as alphafold (<https://alphafold.ebi.ac.uk/>), may highlight any potential differences of the DSS interface in the different species.

Given that the presence of all the 10 subunits in the protomer is essential for proper structure of Dam1 complex, it will be more appropriate to model the entire structure of Dam1 complex instead of the subunits specifically interacting with the DSS. This was not feasible in the present study due to limitations in the computational resources suitable to model this 10-subunit complex. While we do not add information on the DSS neighborhood in other species, we have added biochemical evidence in the revised manuscript that shows that the DSS is essential for the formation of Dam1 rings and its contribution to the Dam1 complex's ability to bind MTs.

5) The basis of the authors claims regarding centromeric chromatin length, ability to bind multiple microtubules and the progressive loss of essentiality of the conserved arginine of Dad2 is not convincing. For example, the authors point out (Fig. 4B legend and Fig. S4A) that *C. neoformans* centromere's ability to bind multiple microtubules is predicted from the size of the centromere, but this is not verified. Furthermore, the authors also point out (page 20 bottom) that they expect at least 10 CENPA nucleosomes to be present on each *C. neoformans* centromere based on the regional centromere length of other species, which may be true, however as mentioned by the authors some large regional centromeres only bind about 2-3 microtubules. Based on the importance of these details for the authors' main argument, these points need to be addressed further and verified experimentally, and not simply "predicted and expected".

We agree to the authors criticism that the binding of multiple kMTs to *C. neoformans* chromosomes are based on predictions. Estimating the number of CENPA nucleosomes by sequencing or microscopic methods are technically challenging and are beyond the main objective of this study. Further, any experiment in this regard requires a strain where the endogenous CENPA is tagged, and it remains functional. In our experience, tagging the only copy of CENPA at its native locus makes it nonfunctional. Nevertheless, we support our argument by analyzing the function of DSS in another yeast, *S. pombe*, known to associate multiple kMTs per chromosome. The similarities in the results from *S. pombe* and *C. neoformans*, considering their phylogenetic distance should lend support to our correlation (Figure 5 in revised manuscript).

6) The experiments performed to investigate the effects of DSS mutants in *C. neoformans* are inadequate for comparing results across the three species studied. For *S. cerevisiae* and *C. albicans*, the authors adequately show that DSS mutants affect cell-cycle progression, spindle dynamics, and timely nuclear segregation. However, they fail to address this for the species they choose to compare these phenotypes with. At the very least, they could have analysed large-budded cells at different temperatures, since this was done for *C. albicans* and showed an increased phenotype at a lower temperature. Further characterization of the mutants in *C. neoformans* in line with what was done for the other species is crucial for a satisfactory comparison.

The phenotype upon depletion of Dam1 complex/other outer kinetochore subunits have been extensively characterized in our previous studies (Kozubowsky et al., *mBio* (2013), Sridhar et al., *Nat. commun* (2020)). Perturbing the Dam1 complex is known to affect chromosome segregation and activate the spindle assembly checkpoint. From the analyses presented in our study (Figure 5B-G in the revised manuscript), we do not find any growth defect or mitotic arrest. Further, our spotting assays suggest that these mutants do not show any sensitivity to thiabendazole or growth at lower temperature. Sensitivities to such stresses that alter the spindle polymerization are a mark of compromised kinetochore-microtubule attachments. The comparable effects upon expression of

CnDad2-FL, CnDad2-R102A, and CnDad2-DSS clearly suggests the redundant nature of DSS/conserved arginine in *C. neoformans*. We have additionally quantified the localization intensity of another Dam1 complex subunit Dad1 to check for the defects in kinetochore integrity upon mutations in the DSS (Figure 5).

7) The authors claim that kinetochore assembly is not affected by the DSS mutations in *C. albicans* and *C. neoformans*, however this is not directly addressed. The authors speculate that this domain may contribute to the overall structure of the DASH/Dam1 complex in *S. cerevisiae*, and since subunits of the DASH/Dam1 complex are not essential in some yeasts, it would be important to know whether these mutants affect the complex structurally. The authors show that the fluorescently labelled dad2-R126A and dad2-R92A mutants in *S. cerevisiae* and *C. albicans* can localize correctly and form foci, but they fail to quantify the fluorescence intensity of the foci, which should be an easy additional analysis for the authors, and especially important since they do show that ScDad2- Δ DSS mutant fluorescence signal is diffused and thus it is possible that the arginine mutants have a lower intensity, which would suggest a compromised kinetochore assembly.

In the revised manuscript, we have included the localization intensity comparison between wild-type and the various mutant versions of Dad2 tested for each species (Figures 1F, 1J, 5F, and 5G) and we do not find statistically significant difference between them.

Figure 1-

Figure 5-

We further investigated the effect of the mutations R126A and DSS deletion on the structure of Dam1 complex in vitro. We find the mutant complexes to be defective in forming rings and are unable to

bind MTs in cosedimentation assays. These findings have been presented as Figure 3 in the revised manuscript (refer to the response to major comment 3 from R1).

In the case of *C. albicans*, any perturbations in the kinetochore ensemble, including perturbing Dad2 itself, is known to result in the disintegration of the entire kinetochore ensemble and subsequent proteasomal degradation of CENPA (Thakur and Sanyal, PLoS Genet (2011)). The kinetochore localization of Dad2-R92A and Dad2-ΔDSS in levels comparable to Dad2-FL is itself indicative of an intact kinetochore ensemble. We also show comparable levels of CENPA between wildtype and mutant strains suggesting that there is no proteasomal degradation of CENPA. This clearly shows that the kinetochore integrity is not compromised in *C. albicans*.

Unlike *C. albicans*, depleting Dam1 complex subunits in *C. neoformans* does not affect the integrity of the kinetochore. Their localization is dependent on other kinetochore proteins of the KMN network or the inner kinetochore. Further, any defects in the outer kinetochore results in a large-bud arrest in this species. (Kozubowsky et al., mBio (2013), Sridhar et al., Nat. Commun (2020)). Comparable levels of wild-type and mutant versions of Dad2, and each of these mutants showing similar Dad1 levels at the kinetochore suggest that kinetochore integrity is not compromised in this species. This is further supported by the lack of mitotic arrest.

Minor points

Representative micrographs of the fluorescence microscopy analyses in Figs. 2C, 3D, 3E, 3H, 3I and 4C, either in the main or supplementary figures would be good.

representative images have been included in the revised manuscript

Page 5, line 8 from bottom: It would help to see the amino acid conservation of the interacting residues from the other DASH/Dam1c proteins within the 'central domain' (see major point 3).

addressed with major point 3.

Page 6, Fig. 1A: The residue numbers above sequence alignment belong to which species?

The numbers in the alignment belongs to *Naumovozyma castellii*. We have mentioned this in the corresponding legend.

Page 7, Fig. 1B legend: the authors should point out here that the structure is based on DASH/Dam1 from another species *Chaetomium thermophilum* as they do in Fig. S3.

We have mentioned this in the legend for figure 1B

Page 8, chapter title and Fig.1 caption: Even though the authors have shown that DSS is important for *S. cerevisiae* Dad2 and less so for *C. albicans*, they have not provided sufficient data in this chapter to conclude that DSS function is "inversely correlated to the centromere length", thus the chapter heading should be adjusted. Similarly, the Fig.1 caption stating that DSS function is "dependent on the length of centromeric chromatin" should be changed (also see major points 2-6).

We have modified the title mentioned above to the following- "The essentiality of the DSS is not a conserved feature across centromere lengths".

Page 9, line 16 from top: The authors state that they "did not observe any significant growth retardation" of ASR03 and ASR06 at 30°C, however it is clear in Fig. 1G that growth is inhibited at 30°C although not to the same extent as 18°C.

Page 9, line 11 from bottom (related to point above): Based on the growth inhibition at 30°C, the authors' conclusions should be adjusted.

This sentence has been modified accordingly in the revised manuscript (P7L) "While we could

observe a mild growth retardation at 30°C, the growth of mutant strains ASR03 and ASR04 expressing CaDad2-R92A and CaDad2-ΔDSS respectively was significantly compromised at a lower temperature of 18°C when compared with CaDad2-FL expressing strain ASR02 or the wild-type strain SN148 grown under similar conditions (Figure 1H)."

Page 10, line 16 from top: Citation missing after "...checkpoint (SAC)"

Citation added

Page 10, line 4 from bottom: The word "colocalized" is not correctly used here. The authors analyse GFP foci between two Spc110-mCherry foci and measure the distance between SPBs and quantify monooriented spindle phenotype. The word is also incorrectly used in figure legends Fig. 2D.

We have modified this phrase in revised manuscript (P8)-“ To validate if the observed defects were consequential of improper kinetochore-MT orientation at metaphase, we studied localization of centromeres (marked by CEN3-GFP) relative to the spindle pole bodies (SPBs, marked by Spc110-mCherry) to monitor the nature of these attachments (Umbreit et al. 2014)."

Page 11, 1st sentence: Reference to Fig. 2D and 2E is missing

Figure citation added

Page 11, lines 3-5 from top: This whole section is unclear and difficult to read

We have simplified this section for better readability

Page 11, Figure 2 legend: The reader does not know what "SPB" is at this stage in the manuscript and should be explained.

We thank the reviewer for pointing this out. We corrected this in the revised version

Page 12, line 17 from bottom: The authors sometime use "CENPA" and sometimes "Cse4" for *C. albicans*, more consistency throughout the manuscript would help readers that are not familiar with the field.

We use the term CENPA to avoid ambiguity in the revised version

Page 12, line 15 from bottom: The authors claim the "kinetochore integrity is not compromised", however this possibility has not been ruled out. For example, quantification of fluorescence intensity of the Dad2-GFP WT and mutants would be beneficial (see major point 7).

Already addressed with major point 7

Page 12, line 13 from bottom: Should be Fig. 1H not 1G. Similar to point above, the authors have not directly investigated kinetochore assembly and further analysis of the kinetochore signal is needed to rule out that deletion of DSS does not affect kinetochore assembly, especially since both the dad2-R92A and dad2-ΔDSS cause growth defects at 30°C and 18°C (Fig. 1G).

We have corrected the figure citation. The comment on kinetochore assembly in these mutants have been addressed in major point 7.

Page 12, the bottom 3 lines: Again, related to the points above, the effects of DSS mutants on kinetochore assembly cannot be ruled out based on the data presented. Furthermore, effects on kinetochore assembly and kinetochore-microtubule interactions are two possibilities that are not mutually exclusive.

already discussed in major point 7.

Page 12, line 3 from bottom: Should be Fig. 1H not 1G.

We have corrected the figure citation.

Page 13, Figure 3 I: The Y axis should be corrected to "unsegregated nucleus" as in Fig. 3C not "stuck nucleus".

The axis label has be corrected

Page 14, line 6 from bottom: The authors state that they observe "increase in the frequency of defective nuclear segregation", however they do not directly show this experimentally, and therefore should use the phrase "unsegregated nuclei" as they do in the figure and figure legends (Fig. 3C).

We have incorporated this term in the revised version

Page 14, line 2 from bottom: Same point as above.

Page 14, line 3 from bottom: "SPB" has not been explained in the main text at this stage in the manuscript.

Already addressed in an earlier comment

Page 15, line 14 from bottom: "mother bud" should be corrected to "mother cell"

We retain the nomenclature mother bud because we do not call it a cell the two buds are separated after cytokinesis/budding.

Page 15, line 4 from bottom: Citation missing after "... C. albicans"

Citation included

Page 16, line 13 from bottom: Reference to Fig. 3H is not needed here.

Figure citation removed

Page 16, line 3 from bottom: What do the authors mean by "wild-type-like behaviour"? This sentence needs more clarification.

This means that the point mutant behaved like wildtype cells in our essays. We have reworded this sentence for clarity- "The tolerance of *S. pombe* and *C. neoformans* to mutations in the DSS is suggestive of a non-essential role for the conserved arginine residue/DSS in Dam1 complex function in these species as compared to *S. cerevisiae* or *C. albicans*"

Page 17, line 2 from top: The authors' conclusion that the "dependence on the conserved arginine residue for mitotic progression across the evolutionary diverged species can be attributed to the differences in the centromeric length" is speculative and the data presented are not strong enough to support this (see major points 2-6).

Discussed in the response to major points

Page 17, line 17 from bottom: The authors do not show any analysis of "rate of chromosome missegregation" in this manuscript, thus this sentence should be removed or modified.

We have rephrased the sentence to "accumulation of cells with unsegregated nucleus"

Page 18, Figure 4C: I'm not sure about LSA policy, but typically new experiments should not be presented for the first time in the discussion, therefore Fig. 4C and the analysis should be relocated to the results section.

Since it is in flow with the discussion thread and there is no objection from the editorial board about this, we retain this panel as it is.

Page 20, line 5-6 from bottom: This sentence is unclear and I'm not sure what the authors mean by "...even in wild-type cells?"

The phrase was used to raise the question on what impact would DSS deletion/ mutation have on cells that naturally have multiple kMTs binding to a chromosome.

Page 21, line 11-12: The sentence is unclear

We have revised the discussion

Page 26, line 3 from bottom: Space missing between "to" and "0.2"

We have made the correction

Page 33, 2nd reference from bottom: Author names are missing.

The reference has been corrected.

Reviewer #3 (Comments to the Authors (Required)):

The study focuses on Dad2 a protein of the ring complex that is essential to tether kinetochores to microtubules. The authors show that a residue of Dad2 R126 is essential in *S.cerevisiae* but not in *Candida albicans* or *cyptococcus neoformans*. They propose that the tolerance is because of regional and not point

kinetochores in *c. albicans* and *c. neoformans*. This is an exciting hypothesis positioning regional centromeres as a fail-safe mechanism which allows tolerance to lethal mutations. The experiments are well done. I would recommend the article for publication with very minor changes.

1. The experiments are well done and the authors present clear evidence for the lack of essentiality of the R128 amino acid in CnDad2. But this is only a correlation with differences in the length of centromeres. So to indicate this clearly, I would recommend the authors clearly double-check their interpretation as a correlation and not as a causal link.

The last sentence of the result (p17) suggests it as a causal link and this should be reworded: "dependence on this conserved arginine residue for mitotic progression across the evolutionarily diverged species can be attributed to the differences in the centromere length."

We have reworded the sentence considering the reviewers comments

2. The discussion and introduction section clearly document that the manuscript is about a single nucleotide in Dad2. However, the abstract opens with a very wide question which could seem to distract the reader from the original findings.

We have modified the abstract and the title of the revised manuscript in line with this suggestion.

3. Fig 3C requires scale bars and mentioning the inversion of intensities

We have made these corrections in figure 3C and its legend.

September 8, 2023

RE: Life Science Alliance Manuscript #LSA-2022-01720-TR

Prof. Kaustuv Sanyal
Jawaharlal Nehru Centre for Advanced Scientific Research
Molecular Biology and Genetics Unit
Bangalore, KA 560064
India

Dear Dr. Sanyal,

Thank you for submitting your revised manuscript entitled "Functional Plasticity in Chromosome-Microtubule Coupling on the Evolutionary Time Scale". We would be happy to publish your paper in Life Science Alliance pending final revisions necessary to meet our formatting guidelines.

- please upload your main manuscript text as an editable doc file
- please upload all figure files as individual ones, including the supplementary figure files; all figure legends should only appear in the main manuscript file
- please upload your Tables in editable .doc or Excel format
- please remove figures from the main manuscript text
- please add your main, supplementary figure, and table legends to the main manuscript text after the references section
- Figure S7 mentions that it was "adapted from Van Hoofe et al. 2017". this Reference is not included in the References list, and it is therefore not clear if permission is needed to adapt this figure here

A. FINAL FILES:

B. MANUSCRIPT ORGANIZATION AND FORMATTING:

Sincerely,

Reviewer #1 (Comments to the Authors (Required)):

The authors of "Functional plasticity in chromosome-microtubule coupling on the evolutionary time scale" have addressed a myriad of concerns raised by this and other reviewers. The concern brought up about testing two models mentioned in the original introduction was sufficiently argued to be not representative of the study design presented in this manuscript. The concerns about adding evolutionary and biochemical confirmation that the DDS motif is important for the Dam1 structure. Indeed, the authors expanded their study to include a new Supplemental Figure S1 and updated Figure 3. Additional concerns raised were adequately addressed. We have no further concerns and recommend the manuscript for publication.

Reviewer #2 (Comments to the Authors (Required)):

I am happy with the revisions and support publication.

Reviewer #3 (Comments to the Authors (Required)):

By analysing over 500 fungi species, the authors identify a new evolutionarily conserved domain in the outer kinetochore protein Dad2 and test its functional significance in models with varied lengths of centromeric DNA. Their biochemical, EM, cell biology and genetic findings together indicate that in the model studies, centromeric DNA offering the binding of multiple kMTs is more robust in masking harmful mutants compared to centromeres offering a single kMT binding site. This is a significant and interesting view on kinetochore plasticity which could be tested in other outer kinetochore proteins and other models. Their interdisciplinary approach along with evolutionary analysis is valuable, and it has revealed a new aspect of kinetochore-microtubule biology.

The authors have satisfactorily addressed the queries raised by the reviewers.

September 19, 2023

RE: Life Science Alliance Manuscript #LSA-2022-01720-TRR

Prof. Kaustuv Sanyal
Jawaharlal Nehru Centre for Advanced Scientific Research
Molecular Biology and Genetics Unit
Molecular Mycology Laboratory
Bangalore, KA 560064
India

Dear Dr. Sanyal,

Thank you for submitting your Research Article entitled "Functional Plasticity in Chromosome-Microtubule Coupling on the Evolutionary Time Scale". It is a pleasure to let you know that your manuscript is now accepted for publication in Life Science Alliance. Congratulations on this interesting work.

DISTRIBUTION OF MATERIALS:

Again, congratulations on a very nice paper. I hope you found the review process to be constructive and are pleased with how the manuscript was handled editorially. We look forward to future exciting submissions from your lab.

Sincerely,
